

# Unveiling the capabilities of vision transformers in sperm morphology analysis: a comparative evaluation

Abdulsamet Aktas[1,2], Gorkem Serbes[3] and Hamza Osman Ilhan[2]

[1] Department of Computer Engineering Faculty of Technology, Marmara University Istanbul, Istanbul, Turkey
[2] Department of Computer Engineering Faculty of Electrical and Electronics, Yildiz Technical University, İstanbul, Turkey
[3] Department of Biomedical Engineering Faculty of Electrical and Electronics, Yildiz Technical University, İstanbul, Turkey

Corresponding author
Abdulsamet Aktas,
abdulsamet.aktas@marmara.edu.tr

## ABSTRACT

Traditional sperm morphology assessment relies on manual visual inspection or semi-automated computer-aided sperm analysis (CASA) systems, which often require labor-intensive pre-processing steps. While recent machine learning approaches, particularly convolutional neural networks (CNNs), have improved feature extraction from sperm images, achieving a fully automated and highly accurate system remains challenging due to the complexity of sperm morphology and the need for specialized image adjustments. This study presents a novel, end-to-end automated sperm morphology analysis framework based on vision transformers (ViTs), which processes raw sperm images from two benchmark datasets-Human Sperm Head Morphology (HuSHeM) and Sperm Morphology Image Data Set (SMIDS)-without manual pre-processing. We conducted an extensive hyperparameter optimization study across eight ViT variants, evaluating learning rates, optimization algorithms, and data augmentation scales. Our experiments demonstrated that data augmentation significantly enhances ViT performance by improving generalization, particularly in limited-data scenarios. A comparative analysis of CNNs, hybrid models, and pure ViTs revealed that transformer-based architectures consistently outperform traditional methods. The BEiT_Base model achieved state-of-the-art accuracies of 92.5% (SMIDS) and 93.52% (HuSHeM), surpassing prior CNN-based approaches by 1.63% and 1.42%, respectively. Statistical significance ($p < 0.05$, $t$-test) confirmed these improvements. Visualization techniques (Attention Maps, Grad-CAM) further validated ViTs' superior ability to capture long-range spatial dependencies and discriminative morphological features, such as head shape and tail integrity. Our work bridges a critical gap in reproductive medicine by delivering a scalable, fully automated solution that eliminates manual intervention while improving diagnostic accuracy. These findings underscore the potential of transformer-based models in clinical andrology, with implications for broader applications in biomedical image analysis.

## INTRODUCTION

Reproduction has always been a crucial aspect of human life. Despite advances, some couples face infertility due to physiological, psychological, or health-related factors. According to the World Health Organization (WHO), infertility is defined as the inability to conceive after one year of unprotected intercourse (*World Health Organization, 2021*), affecting approximately 15% of couples, with male factors accounting for about 50% (*Agarwal et al., 2019*). Identifying male infertility typically begins with a spermiogram, where sperm motility, morphology, and concentration are assessed under a light microscope. This traditional method heavily relies on expert evaluation, making results prone to observer variability and expert fatigue (*Brennan & Silman, 1992*). To address this, computer-aided sperm analysis (CASA) systems have been developed. These combine high-resolution imaging, computational hardware, and image processing software, offering more consistent and accurate results. However, their adoption remains limited due to high costs and strict operational requirements (*Gallagher, Smith & Kirkman-Brown, 2018*).

Sperm motility, concentration, and morphology play an important role in the fertilization of the egg. In particular, sperm with abnormal morphology are directly related to low fertilization and pregnancy rates (*Monte et al., 2013*). Therefore, accurate and consistent assessment of sperm morphology in the spermiogram test is very important in the diagnosis of infertility. According to the WHO, the examination of a minimum of 200 sperm is required to make a highly accurate and reliable diagnosis (*World Health Organization, 1999*). However, visual assessment of 200 sperm per patient, as recommended by the WHO, is highly time-consuming and often impractical in routine clinical settings. As a result, clinicians typically evaluate fewer than 50 sperm, leading to a considerable compromise in diagnostic reliability. More critically, such visual evaluations are inherently subjective and heavily influenced by the expertise, fatigue, and perceptual biases of the observers. This observer variability problem introduces significant inconsistency and inter-observer error, undermining both reproducibility and clinical confidence in manual assessments (*Riddell, Pacey & Whittington, 2005*). Studies have shown that different specialists may assign varying morphology scores to the same sample, making standardized evaluation difficult to achieve. To mitigate these challenges, automated computational approaches have gained significant traction. In particular, machine learning and deep learning-based systems have demonstrated the ability to deliver more consistent, rapid, and cost-effective results compared to manual examination (*Ilhan & Serbes, 2022*). Beyond conventional machine learning and convolutional neural network (CNN)-based models, the recent emergence of vision transformer (ViT) architectures (*Dosovitskiy et al., 2020*) has introduced promising new capabilities for medical imaging applications, including sperm morphology analysis (*Aktas, Serbes & Osman Ilhan, 2023*), owing to their enhanced capacity for modeling long-range dependencies and structural features.

In recent years, several publicly available datasets have been introduced to support the development and evaluation of automated sperm morphology classification methods.

One such dataset is Human Sperm Head Morphology (HuSHeM), created by *Shaker et al., (2017)* consisting of 216 RGB sperm head images with a resolution of 131 ×131 pixels. It includes four morphological classes: normal, pyriform, tapered, and amorphous. While one class represents normal morphology, the remaining three correspond to abnormal types. *Shaker et al. (2017)* applied manual cropping and rotation to standardize sperm orientation, which significantly enhanced the performance of their adaptive patch-based dictionary learning (APDL) model combined with support vector machines (SVM), achieving an accuracy of 92.2%. Although this result is promising, the dependency on manual preprocessing limits its potential for full automation. In addition, the Sperm Morphology Image Data Set (SMIDS), introduced by *Ilhan et al. (2020b)* provides a larger and more diverse dataset comprising approximately 3,000 RGB images with a resolution of 190 × 170 pixels, annotated with three classes: normal, abnormal, and non-sperm. To improve the performance of conventional learning algorithms, an automatic sperm head-tail rotation-based enhancement technique was proposed. Moreover, by combining multiple deep learning architectures and utilizing a two-stage fine-tuning strategy, *Ilhan & Serbes (2022)* achieved the highest reported accuracy on SMIDS to date, reaching 90.87%. The same method yielded a competitive result of 92.1% on the smaller-scale HuSHeM dataset, demonstrating its potential for generalization across datasets (*Ilhan & Serbes, 2022*).

Several studies have investigated sperm morphology classification using the HuSHeM and SMIDS datasets. *Riordon, McCallum & Sinton (2019)* used transfer learning with a pre-trained VGG16 network on ImageNet, followed by fine-tuning for 200 epochs and extensive data augmentation. Although they achieved 94.1% accuracy on HuSHeM, the method required manual image rotation and cropping, limiting its automation potential. *Tortumlu & Ilhan (2020)* evaluated MobileNet-V1 and V2 architectures under varying augmentation conditions; although they obtained 88% accuracy on SMIDS and 77% on HuSHeM, the models were sensitive to augmentation levels and required careful parameter tuning. *Spencer et al. (2022)* combined features of VGG16, VGG19, ResNet-34, and DenseNet-161 using a meta-classifier, reaching 98.2% accuracy on HuSHeM. However, their method also relied on manually rotated images, which, while improving performance, constrained full automation. *Yuzkat, Ilhan & Aydin (2021)* proposed an ensemble of six custom CNNs combined *via* hard and soft voting, reporting 85.18% and 90.73% accuracy for HuSHeM and SMIDS, respectively; still, model complexity and training overhead increased due to multiple architectures. Finally, *Ilhan & Serbes (2022)* introduced a two-stage fine-tuning strategy with VGG-16 and GoogleNet using SMIDS for the adaptation of transfer learning, achieving accuracy of 92.1% and 90.87% in HuSHeM and SMIDS, respectively.

The majority of studies in the literature have used CNN architectures, but transformer-based models are also becoming increasingly popular in the field of medical imaging, showing great promise in various applications. For example, *Benzorgat, Xia & Benzorgat (2024)* proposed an innovative deep learning-based approach for early and accurate brain tumor diagnosis by combining transfer learning and transformer encoder

architecture. The ensemble of DenseNet201, GoogleNet (InceptionV3), and InceptionResNetV2 models was used as feature extractors, which were then fed into a transformer encoder with a shifted window-based self-attention mechanism. The proposed model achieved accuracies of 99.34%, 99.16%, and 98.62% on the Cheng, BT-large-2c, and BT-large-4c datasets, respectively, demonstrating consistent and high performance across different datasets. *Asiri et al. (2024)* also presented a powerful methodology based on the Swin transformer architecture for classifying brain tumors, achieving 97% accuracy. This method outperformed traditional models such as CNN, deep convolutional neural network (DCNN), and ViT, which proved to be a promising approach to accurate tumor detection and classification. Similarly, *Cai et al. (2023a)* introduced the MIST model, based on the Swin transformer, for early differentiation of the malignancy potential of colorectal adenomas. This model, which utilizes self-supervised contrastive learning for feature extraction, achieved a validation accuracy of 78.4%, comparable to local pathologists' performance. Finally, *Cai et al. (2023b)* proposed the Swin Unet3D model, a hybrid approach that combines 3D CNNs for local feature extraction with global context understanding of the vision transformer. The model achieved brain tumor segmentation performance on the BraTS2021 validation dataset with Dice coefficients of 0.840 for the enhancing tumor region, 0.874 for the tumor core and 0.911 for the whole tumor. On the BraTS2018 validation dataset, it obtained Dice coefficients of 0.716 for the enhancing tumor, 0.761 for the tumor core, and 0.874 for the whole tumor. These studies highlight the growing potential and effectiveness of transformer-based models in medical imaging tasks, offering new opportunities for improved clinical diagnosis and treatment planning.

In the field of medical imaging, transformer-based models are becoming increasingly prevalent, and similarly, in sperm morphology classification, transformer architectures have also started to gain popularity. For example, *Mahali et al. (2023)* proposed a hybrid architecture named SwinMobile, which combines the Swin transformer and MobileNetV3, incorporating an autoencoder to denoise the extracted features. They evaluated their model on the HuSHeM, SMIDS, and SVIA datasets, applying data augmentation during both the training and testing phases. Although they reported high accuracies—97.6% on HuSHeM and 91.7% on SMIDS—using augmented samples during the testing introduced bias and compromised generalizability, as it blurred the distinction between training and test sets. Similarly, *Chen et al. (2022)* focused on the SVIA dataset (specifically SubSet-C) to compare vision transformers and CNNs using a balanced subset of sperm and impurity images. Among 12 CNNs and six transformers, DenseNet-121 achieved the best performance with 98.06% accuracy, while ViT yielded the lowest at 91.45%. However, the analysis was limited to a single dataset and two classes, restricting its applicability. Expanding this comparison, *Chen et al. (2023)* added six new transformer models to their benchmark, but continued to rely solely on SubSet-C, maintaining the same limitation. To address generalization more comprehensively, *Aktas, Serbes & Osman Ilhan (2023)* trained five CNN and three transformer models across three datasets—HuSHeM, SMIDS, and SCIAN-Morpho—using consistent 5-fold cross-validation and statistical evaluation with

Student's *t*-test. Each model was compared pairwise in 21 total matches. ViT-L16 emerged as the most effective model with 12 wins, nine draws and no losses in all data sets, while DenseNet-201 performed the worst with zero wins, 13 draws, and eight losses.

Traditional machine learning and deep learning methods have been widely applied to sperm morphology classification, particularly on benchmark datasets such as SMIDS and HuSHeM. However, some studies (*Shaker et al., 2017*; *Riordon, McCallum & Sinton, 2019*; *Spencer et al., 2022*) that report high accuracy rates do not provide fully automated solutions. On the other hand, systems (*Ilhan & Serbes, 2022*; *Ilhan, Serbes & Aydin, 2019*, *2020a*; *Yuzkat, Ilhan & Aydin, 2021*) that offer fully automated solutions with high success rates typically achieve this by combining the outputs of individual learning models. However, these fusion approaches require detailed training of multiple learning models, which results in significantly higher computational costs compared to the training of individual models. ViT models, presented as an alternative to CNNs, have also shown potential to achieve similar or superior performance without the need for ensemble methods (*Aktas, Serbes & Osman Ilhan, 2023*). For a recent example, *Chen et al. (2022*, *2023)* conducted experiments using ViT models with the aim of classifying only two classes of sperm morphology in the SVIA dataset. *Mahali et al. (2023)* applied image augmentation methods to produce new sperm samples and some of these augmented samples were used as test images when model performance was measured. Unfortunately, this practice had caused the production of unreliable learning models in which a dramatic amount of bias had also occurred.

Our literature review revealed that CNNs are predominantly utilized for sperm morphology classification, whereas the application of ViT architectures remains relatively limited. However, recent advances in computer vision have demonstrated that ViTs offer compelling advantages over CNNs, particularly in tasks that require the modeling of long-range dependencies and an enhanced focus on global shape structures rather than localized texture patterns. These characteristics are highly relevant to the domain of sperm morphology, where subtle and complex shape variations must be accurately captured across multiple categories. Motivated by this, our study initially focused on evaluating the performance of several state-of-the-art ViT architectures—traditional ViT (*Dosovitskiy et al., 2020*), BERT pre-training of image transformer (BEiT) (*Bao et al., 2021*), and data efficient image transformer (DEiT) (*Touvron et al., 2021*)—on two publicly available RGB sperm morphology datasets. These models were trained using various optimization algorithms and learning rates as part of a comprehensive hyperparameter sensitivity analysis.

Following this initial analysis, we expanded the scope of our experiments to include widely used CNN-based models and hybrid architectures that combine convolutional and transformer-based components. These models were comparatively evaluated using statistical analysis to provide a more robust performance comparison. Furthermore, we conducted a visual interpretability study to better understand the decision-making mechanisms of the models: Grad-CAM was utilized with CNN architectures, while attention maps were generated for transformer-based models. This qualitative

                                      

assessment revealed key differences in how these architectures attend to morphological regions of interest, thereby offering novel insights into model behavior and highlighting the advantages of ViT models in this context. Collectively, these contributions offer a substantial advancement to the literature by not only benchmarking diverse architectures but also revealing the interpretability and discriminative capacity of each within the scope of sperm morphology classification. This facilitates more dependable and interpretable deep learning models for application in actual clinical decision-support systems.

In the presented study, the limitations and shortcomings of previous approaches have been addressed by the following contributions:

- For the first time in the literature, a diverse range of ViT models were applied to two publicly available multi-class RGB sperm morphology datasets with the aim of identifying the most effective learning architecture.
- A comprehensive hyperparameter analysis procedure was performed, involving various optimization algorithms (SGD, Adamax, RMSprop), learning rates ($10^{-3}$, $10^{-4}$, $10^{-5}$), and data augmentation scales to systematically evaluate model performance under different training conditions.
- In addition to transformer-based models, a set of representative CNN-based architectures and hybrid CNN-transformer models were implemented to provide a broader performance comparison.
- Extensive statistical evaluations were conducted to compare the performance of CNN, transformer and hybrid models in various metrics, ensuring a robust and objective assessment of model effectiveness.
- In order to investigate the internal decision mechanisms of the models, Grad-CAM visualizations were generated for CNN-based models, while attention maps were utilized for transformer-based models. These visual tools identified the specific focus regions utilized by each architecture for sperm morphology classification, providing new interpretability insights in this field.
- The superiority of ViT models over CNNs was demonstrated in detail for multi-class sperm morphology classification, supporting their suitability for modeling morphological variations.

The remainder of this study is organized as follows: 'Materials and Methods' details the materials and methods, describes the datasets, data augmentation techniques, and summarizes the ViT models that achieve the highest accuracies for the sperm morphology datasets in the presented study, along with the proposed approach. 'Experimental Settings' covers the presentation of the experimental results. 'Discussion and Practical Implications' provides a detailed discussion of the findings along with their practical implications in clinical andrology. 'Conclusion' summarizes the conclusions drawn from the study. Finally, 'Limitations and Future Studies' outlines the limitations of the current work and offers directions for future studies.

## MATERIALS AND METHODS

### Experimental setup

To guarantee the reproducibility of the experiment, a precise setup was employed. This section outlines the hardware and software environments used for the experiments. The critical setup details necessary for the replication or extension of this work are summarized in the Supplemental file, which includes specifications of the operating system, hardware, and software libraries.

### Dataset information

HuSHeM was introduced by *Shaker et al. (2017)* for classifying human sperm head shapes. The dataset contains 216 images divided into four classes: normal (54), pyriform (57), tapered (53), and amorphous (52). Images were captured using an Olympus BX50 microscope at 20x magnification. As raw images included tails, multiple heads, and noise, manual pre-processing was applied to crop sperm bodies and rotate heads to the left (*Shaker et al., 2017*; *Riordon, McCallum & Sinton, 2019*). Due to this alignment, *Spencer et al. (2022)* used only vertical flipping for data augmentation. However, such methods are not suitable for developing fully automated systems. In contrast, *Ilhan & Serbes (2022)* trained CNN models directly on raw, unprocessed images and developed a fully automated classification system. Similarly, in our study, models were trained without manual pre-processing. HuSHeM images are stored in RGB format with a resolution of $131 \times 131$ pixels.

The SMIDS dataset was introduced by *Ilhan & Aydin (2019)* using a smartphone-based imaging system. Dataset includes 3,000 labeled images: non-sperm (974), abnormal (1,005), and normal (1,021) (*Ilhan et al., 2020b*). Like HuSHeM, SMIDS is stored in RGB format, with image sizes around $190 \times 170$ pixels. It also includes challenging cases such as noise, mixed tails, and multiple heads.

This study uniquely uses both datasets in their original form, avoiding manual pre-processing steps like cropping or rotation. This approach preserves the data's authenticity and enables more realistic performance evaluation under real-world conditions. HuSHeM and SMIDS were selected as they reflect clinical variability in sperm morphology. Their diversity and inherent imperfections make them robust benchmarks for testing classification models.

### Data augmentation

Preparing a medical image dataset containing sperm samples is challenging due to the microscopic size of regions of interest, requiring expert annotation and careful validation (*Li et al., 2021*). Publicly available sperm morphology datasets with large numbers of labeled images are rare, limiting data availability for deep learning. This scarcity often causes overfitting, as deep models with many parameters—such as CNNs and ViTs—require large datasets to generalize well. *Steiner et al. (2021)* demonstrated that increasing dataset size *via* data augmentation significantly improves ViT model performance. Data augmentation (*Shorten & Khoshgoftaar, 2019*) creates synthetic

samples by applying transformations like flipping, rotation, cropping, and scaling to preserve pattern properties, thereby expanding the training set and enhancing generalization while mitigating overfitting.

In this study, since the sample distribution across classes is balanced, augmentation was applied directly to the training sets without additional class balancing. For HuSHeM, augmentation ranged from 1× (original dataset) to 40×, meaning the dataset size increased 40 times including synthetic samples. For SMIDS, augmentation ratios varied from 1× to 10×. During augmentation, techniques such as flipping, mirroring, and rotation were used to generate diverse synthetic images. Examples of augmented images from SMIDS and HuSHeM are shown in Fig. 1. ViT models were trained on these augmented datasets to analyze how the scale of augmentation affects model performance.

## Proposed approach

CNNs have been widely adopted for image classification due to their strong performance in learning spatial hierarchies (*LeCun et al., 1998*; *Krizhevsky, Sutskever & Hinton, 2012*; *Yamins et al., 2014*). However, their reliance on local receptive fields and weight sharing can limit generalization, frequently leading models to emphasize local textures over global shape. This issue is particularly problematic for tasks dominated by structural features (*Hermann, Chen & Kornblith, 2020*). Some studies have attempted to mitigate these limitations *via* feedback mechanisms to incorporate top-down information (*Cao et al., 2015*), but local connectivity remains a fundamental constraint.

Inspired by their success in modeling global dependencies in NLP (*Devlin et al., 2018*; *Brown et al., 2020*), transformer architectures have recently been adapted for computer vision tasks (*Dosovitskiy et al., 2020*). ViTs process images as patch sequences, allowing them to better capture long-range spatial dependencies and shape-level cues (*Tuli et al., 2021*; *He et al., 2023*). This makes ViTs particularly promising for tasks like sperm morphology classification, where morphological variations (*e.g.*, head size, acrosome structure, tail-body connection) are more critical than texture.

ViTs have shown effectiveness in various medical imaging applications, including COVID-19 diagnosis by X-rays (*Liu & Yin, 2021*; *Than et al., 2021*), and are gaining attention in the analysis of sperm morphology. *Mahali et al. (2023)* proposed a hybrid Swin-Mobile model for sperm head classification, while *Aktas, Serbes & Osman Ilhan (2023)* conducted a comparative study showing the superiority of ViTs over CNNs in this context. Similarly, *Chen et al. (2023)* compared ViTs and CNNs on noisy sperm images, further emphasizing the robustness of ViT-based models.

In comparing CNNs and transformers, two key advantages of ViTs stand out:

- their capability to capture global shape-related dependencies through attention mechanisms
- their structural flexibility to handle varying input sizes without convolution layers (*Tuli et al., 2021*). These features make them well-suited for analyzing morphological differences in sperm cells.

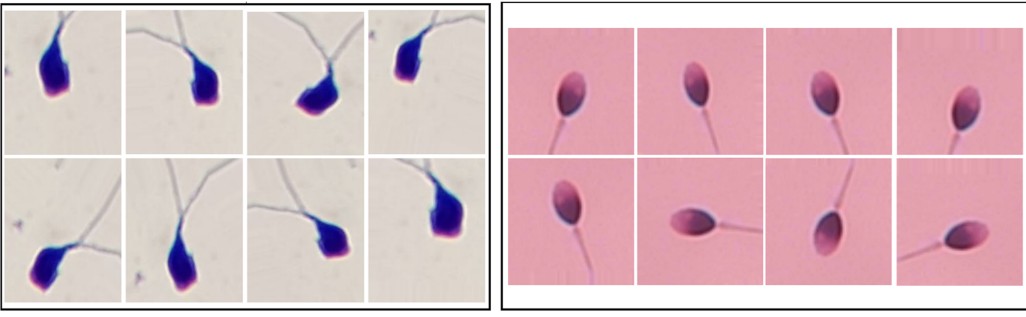

**Figure 1** The image samples obtained by applying data augmentation techniques on SMIDS and HuSHeM (*Shaker et al., 2017*; *Ilhan et al., 2020b*).

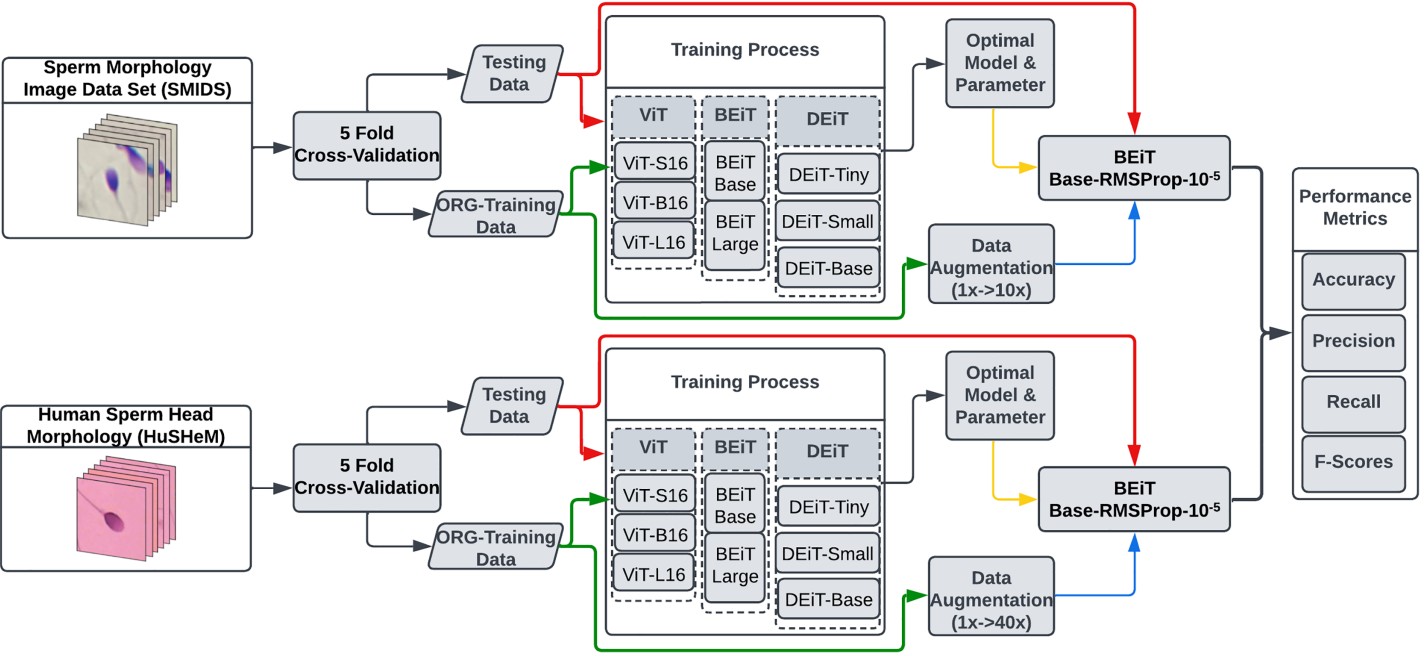

**Figure 2** The flowchart of the proposed transformer based sperm morphology classification approach.

In this study, we explore the potential of eight transformer models for sperm morphology classification using the publicly available HuSHeM and SMIDS datasets. Figure 2 presents the pipeline of our approach. All models were initialized with ImageNet pre-trained weights and fine-tuned using domain-specific samples. To determine optimal configurations, extensive experiments were conducted on data augmentation strategies, optimization techniques, and learning rates under the transfer learning paradigm.

### The ViT model
The ViT, introduced by *Dosovitskiy et al. (2020)*, adapts the transformer architecture from NLP to image classification by dividing input images into patches and encoding their spatial information with positional embeddings. These are passed through a transformer

encoder with multi-head self-attention (MHSA) and a learnable class token for image representation, similar to bidirectional encoder representations from transformers (BERT). The final class prediction is made using an MLP. ViT variants differ by size: ViT-L16 (304 M), ViT-B16 (86.4 M), and ViT-S16 (21 M) parameters. In this study, all three variants were trained on HuSHeM and SMIDS datasets using various optimization strategies and learning rates.

### The data efficient image transformer

ViT models showed strong classification performance on large-scale datasets like JFT-300 M (*Sun et al., 2017*), but their generalization ability depends heavily on data size (*Dosovitskiy et al., 2020*), making them computationally demanding. To overcome this, *Touvron et al. (2021)* proposed the data-efficient image transformer (DEiT), which improves learning efficiency using a novel distillation strategy. DEiT leverages a pre-trained teacher and a randomly initialized student network, introducing a distillation token alongside the class token. This token learns from the teacher's outputs *via* soft or hard-label distillation (*Hinton, Vinyals & Dean, 2015*; *Wei et al., 2020*). In this study, DEiT-Tiny (5 M), DEiT-Small (22 M), and DEiT-Base (86 M) variants were employed.

### The BERT pre-training of image transformer

BERT achieved remarkable success in NLP using the masked language model (MLM) strategy (*Devlin et al., 2018*). Inspired by this, *Bao et al. (2021)* introduced BEiT, a masked image modeling (MIM) approach for pre-training vision transformers. Unlike text, image patches lack sequential structure, so BEiT uses blockwise masking and discrete visual tokens generated by a variational autoencoder (VAE) (*Ramesh et al., 2021*) to restore masked patches. This self-supervised strategy reduces data dependency. In this study, BEiT-Base (86 M) and BEiT-Large (307 M) were used.

A wide range of transformer-based models—including ViT-Small/Base/Large, DeiT-Tiny/Small, and BEiT-Base/Large—were evaluated for sperm morphology classification. These models vary in scale, training methods, and tokenization strategies, allowing for a broad architectural comparison (*Shamshad et al., 2023*; *Jalalifar & Sadeghi-Naini, 2022*; *Aktas et al., 2024*).

After the eight vision transformer variants had been exhaustively tuned (optimizer, learning rate, augmentation scale), a single best-performing recipe emerged—RMSprop with a $10^{-5}$ learning rate and the dataset-specific $40\times/10\times$ augmentation. In the next phase we froze this recipe and applied it, unchanged, to a wider line-up of architectures: state-of-the-art CNN backbones, alternative transformer families, and CNN–transformer hybrids. This design asked a focused question: if every rival model is trained under the very conditions that maximize transformer performance, does any of them bridge the accuracy gap? Five-fold cross-validation and paired *t*-tests answered in the negative—BEiT-base remained significantly ahead—while Grad-CAM (for CNNs) and self-attention maps (for transformers) revealed why: CNNs concentrated on local texture artifacts, whereas transformers attended to the global head-shape contours that define sperm morphology classes. Thus, the "second-round" experiments served as a robustness audit, confirming

**Table 1 ImageNet top-1 Accuracy scores and parameter counts of the utilized models.**

| Models | ViT-S16 | ViT-B16 | ViT-L16 | BEiT-Base | BEiT-Large | DEiT-Tiny | DEiT-Small | DEiT-Base |
|---|---|---|---|---|---|---|---|---|
| ImageNet top-1 Acc. | 80.4 | 84.0 | 85.7 | 83.2 | 85.2 | 72.2 | 79.9 | 81.8 |
| Params | 21M | 86M | 307M | 86M | 307M | 5M | 22M | 86M |

that the observed superiority of transformer architecture is intrinsic rather than a by-product of more diligent tuning.

## Performance metrics

This study comprehensively examines the classification of sperm morphology employing different subvariations of three distinct transformer architectures, each with different hyperparameters. Table 1 presents both the learnable parameter counts for each architecture utilized in this research and their top-1 accuracy on the ImageNet dataset (*Deng et al., 2009*) as documented in the literature.

To assess the performance of the models, accuracy is used as the main metric due to its simplicity and effectiveness in providing a general measure of classification performance. Accuracy, defined as the ratio of correctly predicted samples to the total number of samples in the dataset, is particularly useful for balanced datasets where each class has approximately the same number of samples. In such scenarios, a high accuracy value typically indicates strong overall classification performance.

However, sperm morphology datasets often exhibit class imbalance, where certain morphological categories are underrepresented. In such cases, relying solely on accuracy may lead to misleading interpretations, as a model may achieve high accuracy by predominantly predicting the majority class. To address this limitation and provide a more comprehensive evaluation, additional metrics such as precision, recall, and F1-score were also considered. Precision measures the proportion of true positive predictions among all positive predictions made by the model. It is especially important in medical and biological applications where false positives can lead to unnecessary interventions or misdiagnoses. Recall evaluates the model's ability to correctly identify all actual positive instances. In the context of sperm morphology classification, a high recall ensures that most instances of a particular class (*e.g.*, abnormal morphology) are correctly detected, which is crucial for diagnostic purposes. F1-score, the harmonic mean of precision and recall, provides a single metric that balances both values. It is particularly useful in imbalanced datasets because it is less sensitive to class distribution, offering a more robust measure of performance.

## EXPERIMENTAL SETTINGS

In the proposed study, we developed a system for automated sperm morphology classification using ViT architectures. Eight transformer models, including variations of ViT, BEiT, and DEiT—commonly used in image classification—were evaluated on two datasets. A comprehensive hyper-parameter analysis was conducted to identify the most suitable optimizer, learning rate, and data augmentation scale. For this, three widely used

optimization methods in medical imaging—SGD, RMSprop, and Adamax—were tested (*Sharma et al., 2022*; *Kandel, Castelli & Popovič, 2020*; *Singh et al., 2022*). Each transformer was trained using these optimizers with three learning rates ($10^{-3}$, $10^{-4}$, and $10^{-5}$), resulting in 72 training runs per dataset. Based on these results, the top two transformers per optimizer were selected for further analysis using various data augmentation scales. This analysis aimed to assess not only the effect of augmentation on transformer performance but also their comparative effectiveness against state-of-the-art CNN models. The tested parameter ranges are provided in the Supplemental file TestedHyperparameterList.xlsx.

For a fair comparison, transformer models were evaluated using 5-fold cross-validation, as commonly adopted in the literature (*Ilhan & Serbes, 2022*; *Aktas, Serbes & Osman Ilhan, 2023*; *Ilhan, Serbes & Aydin, 2018a*, *2020a*; *Yuzkat, Ilhan & Aydin, 2021*). The dataset was randomly partitioned into five equal subsets, ensuring uniform sample distribution across folds. In each iteration, 80% of the data was used for training and 20% for testing, allowing all samples to serve as test data once. To avoid initialization bias, transformers were trained from scratch in each fold. Moreover, to ensure consistency across model evaluations, the K-fold partitioning was performed once at the beginning and reused for all architectures. This setup allows an objective and reproducible comparison among transformer models and with prior studies.

The experimental work began by training models on both datasets to find the transformer architecture with the best performance for sperm morphology classification. The optimal hyper-parameter selection procedure started with determining which transformer yielded the best results with which optimization method and learning rate. The two most successful transformers for each optimization method obtained by this selection process were used to examine the impact of data augmentation scale on classification performance.

## EXPERIMENTAL RESULTS

### Determination of the best architecture with optimum hyperparameter configuration

For the SMIDS dataset, a total of 72 model training, consisting of 24 per optimization technique, utilizing eight distinct transformer architectures coupled with three different learning rates. A summary of test accuracy results for these training sessions is presented in Table 2. Adamax, SGD, and RMSProp—three prominent optimization techniques frequently cited in the literature for medical image classification tasks—were assessed. Among these, the Adamax optimizer yielded the highest accuracy. The BEiT Large architecture achieved the best result of 91.33% with a learning rate of $10^{-4}$. The second-best performance, 91.1%, was also obtained by the same architecture at a learning rate of $10^{-5}$, suggesting robustness to changes in learning rate. On the other hand, the results achieved using SGD were not as competitive as those obtained with Adamax. The two most successful results under SGD were obtained with a learning rate of $10^{-3}$. BEiT Large and DEiT Small models achieved accuracies of 89.33% and 86.07%, respectively. In terms of the RMSProp optimizer, the two most successful configurations were achieved

**Table 2** Accuracies of transformer architectures on the original HuSHeM and original SMIDS datasets using Adamax, SGD and RMSProp optimization methods with various learning rates (the best two models for each optimizer are highlighted in bold).

| Dataset | OPT | LR | ViT-S16 | ViT-B16 | ViT-L16 | Beit-Base | Beit-Large | Deit-Tiny | Deit-Small | Deit-Base |
|---------|-----|-----|---------|---------|---------|-----------|------------|-----------|------------|-----------|
| HuSHeM | Adamax | $10^{-3}$ | 70.95 | 71.01 | 84.41 | 60.72 | 64.35 | 63.83 | 59.92 | 64.05 |
| | | $10^{-4}$ | 40.24 | 48.54 | 68.29 | **88.43** | 84.26 | 67.98 | 79.10 | 83.59 |
| | | $10^{-5}$ | 32.33 | 37.50 | 33.21 | 76.39 | **84.72** | 45.16 | 65.81 | 69.72 |
| | SGD | $10^{-3}$ | 52.65 | **56.16** | **74.32** | 40.28 | 47.22 | 39.03 | 37.4 | 22.23 |
| | | $10^{-4}$ | 33.22 | 40.79 | 35.46 | 23.61 | 29.63 | 27.28 | 28.23 | 20.49 |
| | | $10^{-5}$ | 31.82 | 31.92 | 27.24 | 26.85 | 27.31 | 24.61 | 23.64 | 20.00 |
| | RMSprop | $10^{-3}$ | 78.07 | 78.33 | **86.29** | 55.09 | 55.56 | 52.02 | 48.33 | 53.15 |
| | | $10^{-4}$ | 64.61 | 70.64 | 80.24 | 58.80 | 65.78 | 60.23 | 64.58 | 66.30 |
| | | $10^{-5}$ | 38.84 | 39.93 | 47.62 | **88.43** | 84.26 | 71.43 | 78.39 | 78.09 |
| SMIDS | Adamax | $10^{-3}$ | 86.20 | 86.47 | 88.93 | 81.13 | 80.83 | 86.22 | 87.73 | 86.86 |
| | | $10^{-4}$ | 82.33 | 85.06 | 87.23 | 90.77 | **91.33** | 88.83 | 89.83 | 89.46 |
| | | $10^{-5}$ | 69.80 | 71.70 | 78.10 | 90.23 | **91.10** | 87.47 | 87.96 | 88.56 |
| | SGD | $10^{-3}$ | 81.73 | 82.97 | 85.96 | 85.29 | **89.33** | 84.20 | **86.07** | 83.86 |
| | | $10^{-4}$ | 72.20 | 74.03 | 77.66 | 70.96 | 79.80 | 60.36 | 67.33 | 59.93 |
| | | $10^{-5}$ | 48.89 | 48.19 | 50.26 | 40.23 | 34.57 | 35.76 | 49.13 | 35.16 |
| | RMSprop | $10^{-3}$ | 86.26 | 86.47 | 88.36 | 73.43 | 75.70 | 76.23 | 74.07 | 75.56 |
| | | $10^{-4}$ | 86.16 | 86.57 | 88.40 | 81.66 | 82.73 | 86.93 | 87.53 | 89.33 |
| | | $10^{-5}$ | 79.46 | 81.60 | 84.09 | **90.80** | **91.23** | 88.83 | 88.73 | 89.70 |

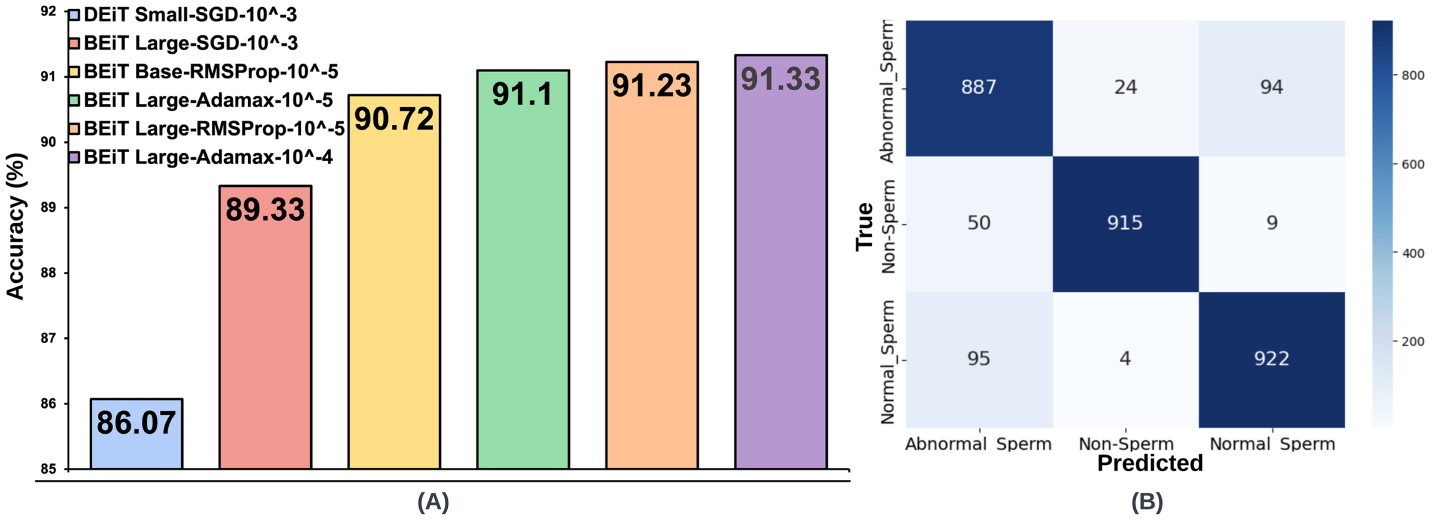

**Figure 3** (A) Optimal hyperparameter configurations and accuracy values for the two best models in each optimization method, (B) Confusion matrix of the BEiT Base-RMSProp-$10^{-5}$ model trained on the ORG-SMIDS.

with a learning rate of $10^{-5}$. BEiT Large was again the top-performing model, reaching an accuracy of 91.23%, closely aligning with its Adamax counterpart. The BEiT Base architecture followed, achieving 90.80% accuracy. The top six architectures, trained with the most optimal hyperparameter configurations across the three optimization algorithms,

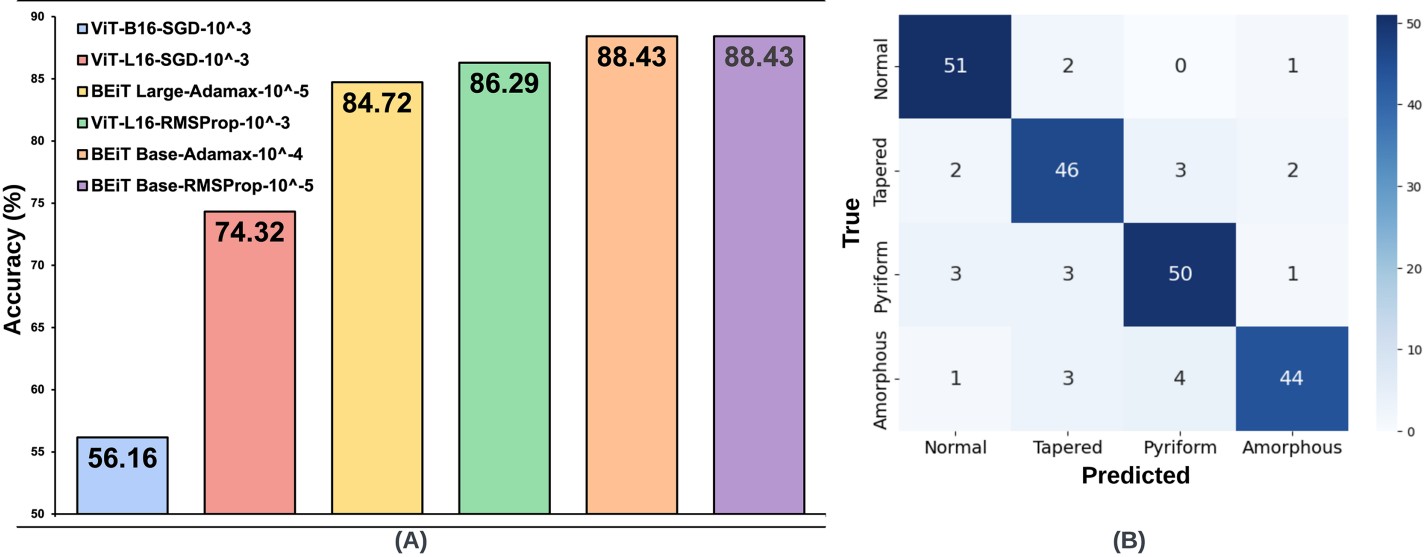

**Figure 4** (A) Optimal hyperparameter configurations and accuracy values for the two best models in each optimization method, (B) Confusion matrix of the BEiT Base-RMSProp-$10^{-5}$ model trained on the ORG-HuSHeM.

are illustrated in Fig. 3. Each architecture in the figure is labeled with its associated optimal hyperparameter settings.

The same processes carried out on the SMIDS were also applied on the HuSHeM. The test accuracies of HuSHeM can be seen in Table 2. According to Adamax results, the highest accuracy value of 88.43% was achieved with BEiT Base architecture using $10^{-4}$ learning rate. This is followed by the BEiT Large model with an accuracy of 84.72% using a learning rate of $10^{-5}$. Similar to the SMIDS, the SGD optimization method was not very successful in the HuSHeM. For the SGD, the most successful result was obtained by ViT-L16 architecture with an accuracy of 74.32% using $10^{-3}$ learning rate. Using the same learning rate, the ViT-B16 architecture is the second most successful models with accuracy values of 56.16%. Although the results obtained with SGD were not very satisfactory, they were used in following steps to examine their relationship with the data augmentation scale. According to the results obtained with the RMSProp, the BEiT Base architecture was the most successful method with an accuracy of 88.43% obtained using a learning rate of $10^{-5}$. With a learning rate of $10^{-3}$, ViT-L16 achieved the second highest accuracy of 86.29%. As in the SMIDS, the results of the six most successful architectures obtained in the HuSHeM can be seen in Fig. 4.

## Evaluation of the best architecture under various augmentation scales

The effect of data augmentation at varying scales on classification performance was investigated by utilizing the hyperparameter settings and model architectures identified during the optimal configuration phase conducted on the original datasets. For the SMIDS dataset, three augmentation levels were applied: 3×, 5×, and 10× the size of the original (ORG) training data. The six architectures previously selected based on optimal performance on SMIDS, as illustrated in Fig. 3, were retrained at each augmentation scale

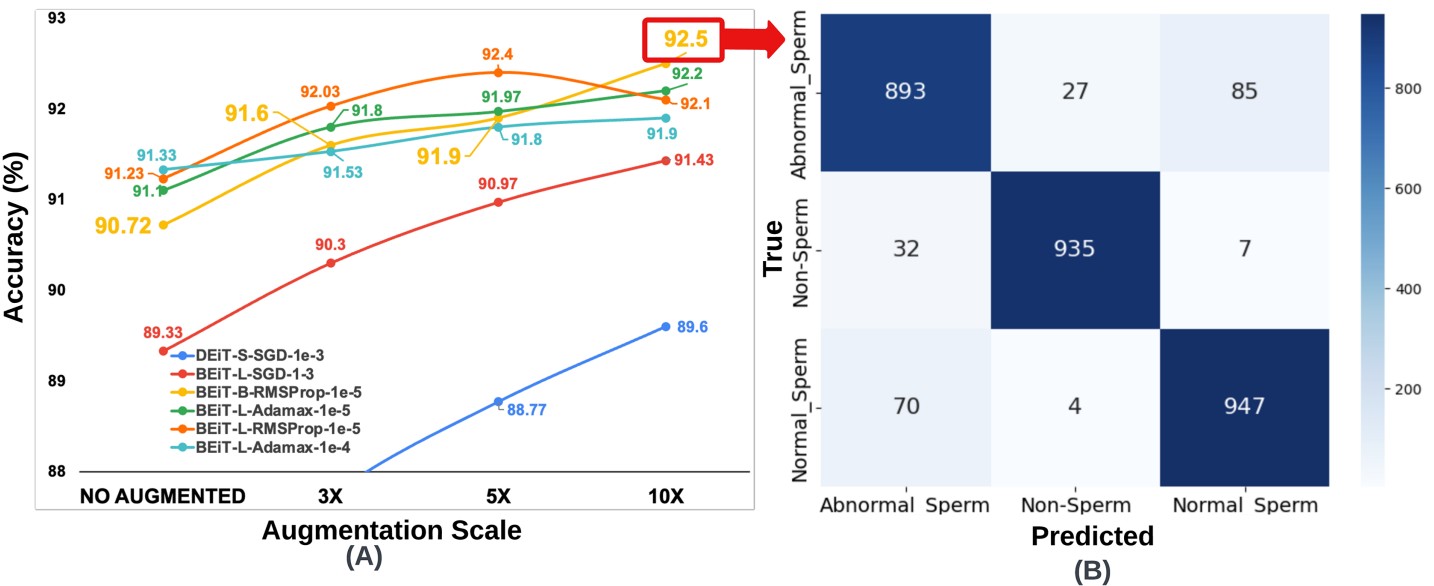

**Figure 5** (A) Classification accuracy values according to data augmentation scales of six architectures obtained through the optimum hyperparameter selection process, (B) Confusion matrix of the BEiT Base-RMSProp-$10^{-5}$ model trained on the 10×-SMIDS.

using their respective best-performing optimization algorithms and learning rates. The same K-fold cross-validation strategy employed in the original training phase was retained for consistency. As a result of the augmentation procedure, three additional training datasets were generated for SMIDS in addition to the original set. Each of the six selected architectures was trained on these augmented datasets using the optimal parameter configurations, yielding a total of 18 additional training processes. To evaluate the impact of augmentation scale, the results obtained from the augmented datasets were systematically compared with those from the original dataset. This comparison serves as an ablation study to assess how data expansion influences model performance, providing insights into the effectiveness of data augmentation in sperm morphology classification. The results obtained for the SMIDS according to the data augmentation scale can be seen in Fig. 5.

To ensure fair comparisons, each learning model was trained for an equal number of epochs (50) by using the same optimization method and learning rate, which were previously set on the training process of the original dataset. The highest accuracy increase for SMIDS was observed in the DEiT Small-SGD-$10^{-3}$ architecture as a jump from 86.07% to 89.6%, while the highest accuracy value of 92.5% was achieved by the BEiT Base-RMSProp-$10^{-5}$ architecture. In almost all models, the success of the learners increased as the scale of data augmentation increased. The only exception of this trend was observed in the BEiT Large-RMSProp-$10^{-5}$ model as a negligible decrease of 0.3% when the augmentation scale changes from 5× to 10×. Regarding to the best performance, highest accuracy values were obtained in almost all learning models at the 10× data augmentation scale with an except for the BEiT Large-RMSProp-$10^{-5}$ architecture, which achieved its

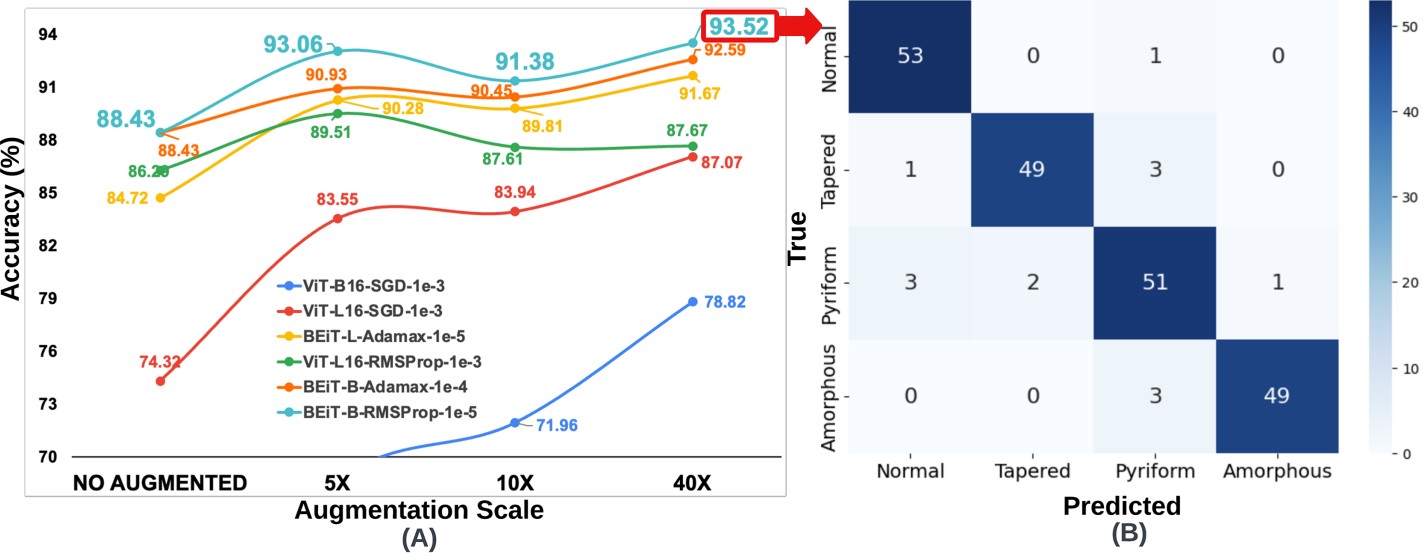

**Figure 6** **(A)** Classification accuracy values according to data augmentation scales of six architectures obtained through the optimum hyperparameter selection process, **(B)** Confusion matrix of the BEiT Base-RMSProp-$10^{-5}$ model trained on the 40×-HuSHeM.

highest value of 92.4% accuracy at the 5× data augmentation scale as shown in Fig. 5. Comparing the performance of two most successful learning models, it was seen that there is only a 0.1% difference between the BEiT Large-RMSProp-$10^{-5}$ architecture and the Beit Base-RMSProp-$10^{-5}$ architecture, when the BEiT Large-RMSProp-$10^{-5}$ architecture was applied to 5× data augmentation scale, while the BEiT Base-RMSProp-$10^{-5}$ architecture was applied to 10× data augmentation scale, respectively. However, comparing the model complexity, the BEiT Large-RMSProp-$10^{-5}$ architecture has 307M parameters while the BEiT Base-RMSProp-$10^{-5}$ architecture has only 86M parameters. Therefore, it would be more appropriate to choose the Beit Base-RMSProp-$10^{-5}$ architecture for the classification of sperm morphology in SMIDS.

Since the number of samples in the HuSHeM is fewer than in the SMIDS, a range of larger data augmentation scales were applied. The HuSHeM was increased up to 40× of the original dataset. For the HuSHeM, according to the findings of parameter selection process, six selected architectures were trained on data augmented data-sets just as performed in the SMIDS. The K-fold cross-validation approach used to train the original dataset was also applied here. In addition to the original architectures, the results of 18 training processes for architectures trained at different data augmentation scales can be seen in Fig. 6. Likewise the SMIDS training procedure, each learning model was trained for an equal number of epochs by using the same optimization method and learning rate, which were previously set on the training process of the original dataset. When analyzing the results, it is realized that the highest accuracy increase is observed in ViT-B16-SGD-$10^{-3}$ architecture with 22.66%. It should be noted that only 56.16% accuracy was obtained on the original dataset, while 78.82% accuracy was achieved on the 40× dataset by using the same parameters. However, the highest success value was obtained by the BEiT

**Table 3** The evaluation of BEiT Base-RMSProp-$10^{-5}$ in terms of confusion matrix based performance metrics derived from Figs. 3, 4, 5 and 6.

| | | HuSHeM | | | | SMIDS | | |
|---|---|---|---|---|---|---|---|---|
| | | Normal | Tapered | Pyriform | Amorphous | Normal | Abnormal | Non-Sperm |
| Original dataset | Precision | 89% | 85% | 87% | 92% | 86% | 97% | 90% |
| | Recall | 94% | 87% | 88% | 85% | 88% | 95% | 90% |
| | F1-score | 92% | 86% | 88% | 88% | 87% | 95% | 90% |
| | Accuracy | 88.43% | | | | 90.80% | | |
| Augmented dataset (40×–10×) | Precision | 91% | 96% | 88% | 98% | 90% | 97% | 91% |
| | Recall | 98% | 92% | 89% | 94% | 89% | 96% | 93% |
| | F1-score | 94% | 94% | 88% | 96% | 89% | 96% | 92% |
| | Accuracy | 93.52% | | | | 92.5% | | |

Base-RMSProp-$10^{-5}$ architecture, which reached 93.52% accuracy with an increase of 5.09% compared to the original dataset. In the HuSHeM, as in the SMIDS, the success of the architectures tends to increase as the scale of data augmentation increases as demonstrated in Fig. 6.

When analyzing the results obtained on both datasets, it is seen that the BEiT Base-RMSProp-$10^{-5}$ architecture is the most successful model for sperm morphology classification. Table 3 was created using the confusion matrices given in Figs. 3, 4, 5 and 6. In Table 3; the precision, recall and F1-score values of the models, which were trained with the BEiT Base-RMSProp-$10^{-5}$ architecture both on the original dataset, and on both 10× data augmented SMIDS and 40× augmented HuSHeM, are presented. When comparing the precision scores, also called positive prediction value, Beit Base architecture trained with the most optimal optimization method and learning rate has performed well on both datasets. In HuSHeM, the highest increase was obtained in the abnormal type of Tapered with 11%, while in SMIDS, the highest increase was obtained in the Normal class with 4% when using augmented data. A promising improvement of precision score was also achieved for the Amorphous abnormal type in the HuSHeM. When analyzing the difference in recall scores between original and augmented tests, the false negative values for the Normal class were significantly reduced resulting in achieving the highest recall value for HuSHeM. On the other hand, the highest increase in recall value was obtained in Amorphous class with 9%. Regarding the SMIDS, an increase in recall values was observed in all classes. When F1-scores were analyzed, a significant increase was obtained in all classes except Pyriform class in HuSHeM. In SMIDS, an F1-score increase was achieved in all classes. Overall accuracy values show an increase of 5.09% for HuSHeM and 1.7% for SMIDS using augmented data. In the light of the obtained results, it is observed that the scale of data augmentation has a positive effect on model success.

### Cross-architecture benchmarking under a unified training recipe

To verify that the advantage obtained with the BEiT-Base transformer is intrinsic rather than a by-product of architecture-specific tuning, we froze the training recipe that proved optimal in the transformer sweep—RMSprop (learning rate $10^{-5}$) together with the

dataset-specific 40× (HuSHeM) and 10× (SMIDS) augmentation—and retrained a broader palette of networks under *exactly* the same conditions. This palette comprised modern CNN backbones (a lightweight custom CNN and EfficientNet-V2-S/L) (*Tan & Le, 2021*), hierarchical transformers that embed convolutional priors (Swin-V2-T/L) (*Liu et al., 2022*), and two flavours of CNN–transformer fusion: hand-crafted cascades that route CNN feature maps into a transformer head, and the unified CoAtNet-0/1 architecture (*Dai et al., 2021*), which interleaves depth-wise convolution with self-attention in a single backbone. By enforcing a common optimization schedule and data regime across all candidates, we removed training-procedure bias, enabling a fair test of whether convolutional or hybrid designs can close the margin set by the best pure transformer; the results therefore reflect differences in representational capacity rather than differences in hyperparameter diligence.

### Convolutional baselines

To establish a texture–centric point of comparison we first trained three pure-CNN models. (1) A lightweight custom CNN with five convolutional blocks (1.3 M parameters) serves as a lower-capacity reference and helps gauge the benefit of transfer learning. (2)–(3) EfficientNetV-2-S and EfficientNet-V2-L embody state-of-the-art scaling principles; they achieve competitive ImageNet accuracy and therefore reveal how far carefully engineered convolutions alone can be pushed when they receive the exact same optimiser, learning-rate, and augmentation schedule that was found optimal for the transformers (*Tan & Le, 2021*). The EfficientNet-V2-S model is designed to operate with low resource requirements and demonstrates high efficiency, particularly with small-scale datasets. On the other hand, the EfficientNet-V2-L model has a larger number of parameters and a deeper architecture.

### Alternative transformer baselines

The ViT analysis pinpointed BEiT-Base as the most robust single-backbone model, thus it serves as the reference transformer in subsequent sections. To test whether the observed advantage is unique to BEiT or extends to other attention-centric designs, we additionally include Swin transformer V2 in its Tiny and Large variants (*Liu et al., 2022*). Swin V2 introduces shifted local windows and hierarchical feature maps, thereby injecting a convolution-like inductive bias while preserving global self-attention; its presence therefore probes whether architectural hybrids inside a single backbone can rival a plain ViT derivative under identical training conditions.

### CNN-transformer fusion (hybrid) models

Beyond single-backbone networks, two explicit fusion technique were examined. Cascade models pass convolutional feature volumes from either the Custom CNN or EfficientNetV-2-S into a BEiT-Base or Swin V2-L head, testing whether a staged pipeline can combine the locality of CNNs with the long-range modelling of transformers. Complementing these hand-crafted cascades, CoAtNet-0/1 embeds depthwise convolutions and self-attention within each stage of a unified backbone, providing an "off-the-shelf" fusion engineered by Google Research that has shown strong performance on

**Table 4  5-Fold cross-validation accuracy results of CNN-, Transformer-, and Hybrid-based models on the 40-x HuSHeM (the best performing model is highlighted in bold).**

**HuSHeM40x**

| Methods | Model name | Fold 1 | Fold 2 | Fold 3 | Fold 4 | Fold 5 | Avg. |
|---|---|---|---|---|---|---|---|
| CNN based network | Custom CNN | 68.89% | 62.22% | 65.12% | 78.57% | 92.68% | 73.50% |
| | Eff-V2-S | 82.22% | 86.67% | 93.02% | 100.00% | 90.24% | 90.43% |
| | Eff-V2-L | 86.67% | 80.00% | 93.02% | 100.00% | 90.24% | 89.99% |
| Transformer based network | **BEiT-B** | **93.33%** | **88.70%** | **95.35%** | **100.00%** | **90.24%** | **93.52%** |
| | Swin-V2-L | 86.67% | 84.44% | 90.70% | 100.00% | 92.68% | 90.90% |
| | Swin-V2-T | 88.89% | 84.44% | 93.02% | 100.00% | 87.80% | 90.83% |
| Hybrid network | Custom CNN + BEiT-B | 86.67% | 82.22% | 90.70% | 100.00% | 90.24% | 89.97% |
| | Eff-V2-S + BEiT-B | 80.00% | 82.22% | 95.35% | 100.00% | 90.24% | 89.56% |
| | Eff-V2-S + Swin-V2-L | 82.22% | 82.22% | 90.70% | 100.00% | 87.80% | 88.59% |
| | CoAtNet-0 | 80.00% | 77.78% | 88.37% | 100.00% | 87.80% | 86.79% |
| | CoAtNet-1 | 86.67% | 80.00% | 90.70% | 100.00% | 85.37% | 88.55% |

large-scale vision benchmarks (*Dai et al., 2021*). The CoAtNet architecture offers both efficient local feature extraction capabilities of convolutional neural networks and the ability to model long-range dependencies, as provided by transformer-based models.

All architectural models evaluated in this study were optimized using the RMSprop algorithm with a fixed learning rate of $10^{-5}$ across 50 training epochs. The computational characteristics of each model, including parameter counts, average training time per epoch, and total training time for 50 epochs across five folds, are systematically documented in the Supplemental file TrainingTimeArchitectures.

### Cross-architecture performance and statistical analysis

Table 4 summarizes the 5-fold cross-validation performance metrics for CNN-, transformer-, and hybrid-based architectures evaluated on the $40\times$ HuSHeM dataset. Our experimental results demonstrate that the transformer-based BEiT-B architecture achieved state-of-the-art performance, attaining a mean classification accuracy of 93.52% with a standard deviation of 4.61%, which significantly outperformed all other models ($p < 0.05$, paired $t$-test). The Swin transformer V2 variants (Swin-V2-L: $90.90 \pm 5.57\%$; Swin-V2-T: $90.83 \pm 5.77\%$) also exhibited strong competitive performance, confirming the effectiveness of hierarchical transformer architectures for this task.

Among CNN-based approaches, the EfficientNetV2 family achieved consistent performance with average accuracies of $90.43 \pm 5.55\%$ for Eff-V2-S and $89.99 \pm 6.57\%$ for Eff-V2-L, while our custom CNN implementation showed markedly lower accuracy ($73.50 \pm 11.90\%$), highlighting the advantages of transfer learning and architectural optimization in modern deep networks. The hybrid architectures, particularly the Custom CNN + BEiT-B combination ($89.97 \pm 6.30\%$), demonstrated that integration of convolutional and attention mechanisms can achieve competitive accuracy, though they did not surpass pure transformer-based solutions in performance. This suggests that while hybrid approaches

**Table 5  5-Fold cross-validation accuracy results of CNN-, Transformer-, and Hybrid-based models on the 10-x SMIDS  (the best performing model is highlighted in bold).**

**SMIDS10x**

| Methods | Model name | Fold 1 | Fold 2 | Fold 3 | Fold 4 | Fold 5 | Avg. |
|---|---|---|---|---|---|---|---|
| CNN based network | Custom CNN | 78.13% | 79.70% | 80.00% | 80.00% | 82.17% | 80.00% |
| | Eff-V2-S | 90.65% | 89.85% | 92.00% | 91.33% | 91.83% | 91.13% |
| | Eff-V2-L | 91.99% | 90.35% | 92.33% | 88.83% | 91.83% | 91.07% |
| Transformer based network | **BEiT-B** | **92.65%** | **92.51%** | **92.33%** | **92.00%** | **93.00%** | **92.50%** |
| | Swin-V2-L | 92.15% | 91.68% | 91.50% | 90.17% | 92.83% | 91.67% |
| | Swin-V2-T | 91.82% | 91.01% | 91.17% | 89.33% | 92.33% | 91.13% |
| Hybrid network | Custom CNN + BEiT-B | 91.32% | 91.68% | 91.67% | 90.17% | 93.00% | 91.57% |
| | Eff-V2-S + BEiT-B | 91.15% | 92.01% | 90.67% | 87.67% | 91.83% | 90.67% |
| | Eff-V2-S + Swin-V2-L | 91.15% | 90.35% | 91.83% | 91.00% | 91.50% | 91.17% |
| | CoAtNet-0 | 91.49% | 90.18% | 91.17% | 89.00% | 91.83% | 90.73% |
| | CoAtNet-1 | 91.15% | 90.85% | 91.33% | 89.50% | 92.50% | 91.07% |

provide an interesting architectural alternative, transformer-based models currently offer superior classification capability for this specific morphological analysis task.

Table 5 summarizes the 5-fold cross-validation results on the $10\times$ augmented SMIDS dataset. Experimental findings reveal that the transformer-based BEiT-B model achieved statistically significant superior performance ($p < 0.05$, paired $t$-test) with 92.50% mean accuracy ($\sigma = 0.34\%$), demonstrating both the highest accuracy and lowest variance. The hierarchical transformer architectures, Swin-V2-L ($91.67 \pm 1.01\%$) and Swin-V2-T ($91.13 \pm 1.17\%$), also exhibited notable performance. Among CNN-based models, EfficientNetV2-S ($91.13 \pm 0.78\%$) and EfficientNetV2-L ($91.07 \pm 1.19\%$) achieved high accuracy values benefiting from pretrained weights, while the custom CNN model ($80.00 \pm 1.43\%$) showed significantly lower performance. The hybrid architectures, particularly the Custom CNN + BEiT-B ($91.57 \pm 1.01\%$) and EfficientNetV2-S + Swin-V2-L ($91.17 \pm 0.55\%$) combinations, demonstrated the potential of multi-architectural approaches, though they couldn't match the performance level of pure transformer models.

To evaluate whether the observed differences in classification performance across models were statistically significant, we conducted pairwise comparisons using a two-tailed paired $t$-test at a significance level of $\alpha = 0.05$. The tests were performed on accuracy scores obtained from 5-fold cross-validation across two distinct datasets. Table 6 presents the results of the paired $t$-test conducted on the performance scores obtained *via* 5-fold cross-validation, along with the number of wins, draws, and losses (W/D/L) observed in the model comparisons. Given that the comparisons are conducted on two distinct datasets, the total number of win/draw/loss (W/D/L) outcomes for each model comparison amounts to two. A statistically significant difference was considered to exist when the $p$-value was less than 0.05. The BEiT-Base model outperformed all other models with 12 statistically significant wins and eight draws ($p < 0.05$ in 12 pairwise tests), showing no losses across comparisons—highlighting its robustness and consistent superiority.

**Table 6 Paired *t*-test analysis of deep learning models based on CNN, Transformer, and Hybrid architectures over 5-fold cross-validation (the best performing model is highlighted in bold).**

| | CNN | Eff-V2-S | Eff-V2-L | Beit-B | Swin-V2-L | Swin-V2-T | Custom CNN + BEiT-B | Eff-V2-S + BEiT-B | Eff-V2-S + Swin-V2-L | CoAt-0 | CoAt-1 |
|---|---|---|---|---|---|---|---|---|---|---|---|
| Custom CNN | – | 2/0/0 | 2/0/0 | 2/0/0 | 2/0/0 | 2/0/0 | 2/0/0 | 2/0/0 | 2/0/0 | 2/0/0 | 2/0/0 |
| Eff-V2-S | 0/0/2 | – | 0/2/0 | 1/1/0 | 0/2/0 | 0/2/0 | 0/2/0 | 0/2/0 | 0/2/0 | 0/2/0 | 0/2/0 |
| Eff-V2-L | 0/0/2 | 0/2/0 | – | 0/2/0 | 0/2/0 | 0/2/0 | 0/2/0 | 0/2/0 | 0/2/0 | 0/1/1 | 0/2/0 |
| BEiT-B | 0/0/2 | 0/1/1 | 0/2/0 | – | 0/1/1 | 0/0/2 | 0/1/1 | 0/1/1 | 0/1/1 | 0/1/1 | 0/0/2 |
| Swin-V2-L | 0/0/2 | 0/2/0 | 0/2/0 | 1/1/0 | – | 0/1/1 | 0/2/0 | 0/2/0 | 0/2/0 | 0/1/1 | 0/1/1 |
| Swin-V2-T | 0/0/2 | 0/2/0 | 0/2/0 | 2/0/0 | 1/1/0 | – | 0/2/0 | 0/2/0 | 0/2/0 | 0/1/1 | 0/1/1 |
| Custom CNN + BEiT-B | 0/0/2 | 0/2/0 | 0/2/0 | 1/1/0 | 0/2/0 | 0/2/0 | – | 0/2/0 | 0/2/0 | 0/1/1 | 0/1/1 |
| Eff-V2-S + BEiT-B | 0/0/2 | 0/2/0 | 0/2/0 | 1/1/0 | 0/2/0 | 0/2/0 | 0/2/0 | – | 0/2/0 | 0/2/0 | 0/2/0 |
| Eff-V2-S + Swin-V2-L | 0/0/2 | 0/2/0 | 0/2/0 | 1/1/0 | 0/2/0 | 0/2/0 | 0/2/0 | 0/2/0 | – | 0/2/0 | 0/2/0 |
| CoAtNet-0 | 0/0/2 | 0/2/0 | 1/1/0 | 1/1/0 | 1/1/0 | 1/1/0 | 1/1/0 | 0/2/0 | 0/2/0 | – | 0/2/0 |
| CoAtNet-1 | 0/0/2 | 0/2/0 | 0/2/0 | 2/0/0 | 1/1/0 | 1/1/0 | 1/1/0 | 0/2/0 | 0/2/0 | 0/2/0 | – |
| Win/Draw/Lose | 0/0/20 | 2/17/0 | 3/16/0 | **12/8/0** | 5/14/1 | 4/13/3 | 4/15/1 | 2/17/1 | 2/17/1 | 2/13/5 | 2/13/5 |

EfficientNet-V2 variants also performed well but did not achieve as many statistically significant victories. Conversely, some hybrid and CoAtNet-based models were statistically outperformed in several comparisons, reflected by their higher loss counts. These findings emphasize the effectiveness of transformer-based architectures, particularly BEiT-Base, in capturing morphological features relevant to sperm classification, and highlight the importance of statistical testing in model evaluation beyond raw performance metrics.

In order to better understand the decision-making processes of the evaluated architectures, a qualitative interpretability analysis was also conducted using visual explanation techniques. In this context, the Grad-CAM method was employed for the EfficientNet-V2 model, while self-attention maps were utilized for the BEiT-Base transformer model. As shown in Figs. 7 and 8, Grad-CAM and attention map visualizations for eight different sperm cell samples selected from the SMIDS and HuSHeM datasets are presented. Each image includes the ground-truth class labels and the corresponding predictions made by each model. To gain insight into why transformer-based models outperform CNN-based models in classification tasks, particular attention was given to samples where the EfficientNet-V2 model failed but the BEiT-Base model succeeded in classification.

Figure 7 displays visualizations for samples from the HuSHeM dataset, covering four different classes (Amorphous, Pyriform, Tapered, and Normal). Samples 1 through 4 belong to the Amorphous class, which includes sperm cells with no distinct morphological structure, with abnormalities potentially affecting the head, neck, or both regions. The Grad-CAM outputs of EfficientNet-V2 show that the model primarily focuses on the head region and fails to evaluate the sperm cell holistically. In contrast, the attention maps from the BEiT-Base model indicate a more comprehensive analysis of the entire sperm cell, leading to more accurate classifications.

Samples 5, 6, and 7 belong to the Pyriform class, characterized by sperm cells with a relatively normal upper part and a significantly narrower lower part. The visualizations

**Figure 7 Interpretability analysis of deep learning models for sperm morphology classification: a comparative visualization of BEiT-base attention maps and EfficientNet-V2 Grad-CAM on the HuSHeM dataset (*Shaker et al., 2017*).**

reveal that the transformer model typically focuses on the lower section of the sperm, particularly the junction with the tail, whereas the CNN model directs its attention more broadly or only partially to this region. This focused attention contributes to the superior decision-making performance of the transformer model. Sample 8 represents the Tapered class, in which the sperm cell appears laterally compressed. In this case, the BEiT-Base model effectively attends to the top, bottom, and lateral areas of the cell, while the EfficientNet-V2 model shows limited activation confined to a narrow region, resulting in a misclassification.

Similar analyses were conducted for the SMIDS dataset and are presented in Fig. 8. Samples 1, 2, and 3 belong to the Abnormal class. In these examples, the BEiT-Base model

**Figure 8 Interpretability analysis of deep learning models for sperm morphology classification: a comparative visualization of BEiT-base attention maps and EfficientNet-V2 Grad-CAM on the SMIDS dataset (*Ilhan et al., 2020b*).**

effectively focuses on the head region, capturing class-specific morphological features, whereas the EfficientNet-V2 model tends to concentrate on more general and less discriminative areas. Samples 6, 7, and 8 belong to the Normal class, where sperm cells are located in close proximity to artifacts. The EfficientNet-V2 model, misled by these artifacts, incorrectly classifies the images as non-sperm. In contrast, the BEiT-Base model is unaffected by such noise and correctly focuses on the relevant sperm morphology. Sample 5 represents a non-sperm class instance. While the EfficientNet-V2 model mistakenly interprets dye remnants as indicative of a sperm cell, the BEiT-Base model correctly identifies the absence of any meaningful sperm-related structures and classifies the sample accordingly.

Overall, these findings demonstrate that the transformer-based BEiT-Base model not only achieves higher classification accuracy but also exhibits more meaningful and anatomically coherent attention distributions during decision-making processes. Especially in the context of identifying morphological abnormalities, the model's ability to holistically assess sperm cells leads to more reliable and explainable outcomes compared to the CNN-based EfficientNet-V2. This highlights the advantages of transformer architectures not only in terms of performance but also in interpretability and clinical reliability.

## DISCUSSION AND PRACTICAL IMPLICATIONS

Sperm morphology analysis plays a crucial role in the diagnosis of male infertility. Traditionally, two methods are used in clinical practice: visual assessment by specialists and CASA systems. While visual assessment is widely practiced, it is inherently subjective and prone to inter-observer variability due to the dependence on specialist expertise. CASA systems offer a more consistent and objective alternative, but most existing systems lack full automation, especially in morphology evaluation. This study addresses these limitations by introducing a fully automated, transformer-based analysis framework with enhanced classification accuracy.

Unlike traditional CNN-based models that are often texture-focused, transformer architectures are known to emphasize shape-related features, which are highly relevant in sperm morphology classification where head shape abnormalities are the primary discriminative feature. As demonstrated in previous work (*Hermann, Chen & Kornblith, 2020*), transformers outperform CNNs in tasks requiring shape sensitivity. By leveraging this property, the proposed framework capitalizes on the shape-dependent nature of the problem.

Extensive experiments were conducted using two benchmark datasets: HuSHeM and SMIDS. These datasets differ in scale and quality, allowing for a comprehensive evaluation of the model's generalization capacity. The BEiT Base transformer, trained with the RMSProp optimizer and a learning rate of $10^{-5}$, achieved state-of-the-art performance with 93.52% accuracy on HuSHeM and 92.5% on SMIDS, outperforming previous fully automated solutions. A detailed comparison is provided in Table 7.

Another important factor influencing model success is the choice of training hyperparameters and the scale of data augmentation. In this study, different augmentation levels were systematically evaluated, and optimal training configurations were identified. The results underscore the importance of careful training strategy design for maximizing the capabilities of data-hungry transformer architectures.

### Practical deployment considerations

For the proposed system to be adopted in clinical or research settings, several practical considerations must be addressed:

- **Full automation:** The proposed pipeline operates without manual pre-processing steps such as cropping or rotation, making it highly suitable for clinical integration. This ensures reproducibility and efficiency in high-throughput lab environments.

**Table 7 Comparison of sperm morphology classification between the proposed approach and approaches in the literature.**

| Data set name | Data set information | Orientation and cropping | Article | Method | Accuracy (%) |
|---|---|---|---|---|---|
| HuSHeM | 216 sperm images 131 × 131 resolution overlapped four sperm classes | Manual | *Shaker et al. (2017)* | Manually Sperm Head Rotation and cropping into 50 × 76 pixels Color Space Converting Dictionary Learning | 92.2 |
| | | Automatic | *Ilhan, Serbes & Aydin (2019)* | Automatic Directional Masking k-NN classification | 57.4 |
| | | Manual | *Riordon, McCallum & Sinton (2019)* | Data Augmentation Transfer Learning using VGG16 (ImageNet) | 94 |
| | | Automatic | *Ilhan, Serbes & Aydin (2020a)* | Adaptive De-Noising Overlapping Group Shrinkage Image Gradient Directional Masking Feature Extraction by MSER descriptor SVM Classification (non-linear Kernel) | 86.6 |
| | | Not applied | *Yuzkat, Ilhan & Aydin (2021)* | Data Augmentation Transfer Learning using CNN Architectures Decision Level Fusion | 85.18 |
| | | Not applied | *Ilhan & Serbes (2022)* | Data Augmentation Transfer Learning using VGG16 and GoogleNet (ImageNet + SMIDS) Decision Level Fusion (VGG16 + GoogleNet) | 92.13 |
| | | Semi-manual | *Spencer et al. (2022)* | Data Augmentation Transfer Learning VGG16, VGG19, Resnet-34 and DenseNet-161 Meta-classifier | 98.2 |
| | | Not applied | *Mahali et al. (2023)* | Inappropriate Data Augmentation Transfer Learning SwinTransformer, MobileNetV3 and Auto encoder Fusion models (SwinMobile) | 97.6 |
| | | Not applied | *Aktas, Serbes & Osman Ilhan (2023)* | Data Augmentation Transfer Learning CNNs and Transformers | 90.85 |
| | | Not applied | Proposed study 2024 | Data Augmentation Transfer Learning ViTs, DEiTs and BEiTs | 93.52 |
| SMIDS | 536 sperm patches 190 × 170 resolution overlapped three classes | Not applied | *Ilhan, Serbes & Aydin (2018a)* | Dual Tree Complex Wavelet Trans. SVM classification | 82.3 |
| | | | *Ilhan et al. (2018b)* | Adaptive De-Noising Feature Extraction by SURF Descriptor SVM classification | 83.4 |
| | | Not applied | *Ilhan et al. (2020b)* | Overlapping Group Shrinkage Fuzzy-C Means Clustering Data Augmentation Classification by MobileNet | 87 |
| | 3,000 sperm patches 190 × 170 resolution overlapped three classes | Automatic | *Ilhan, Serbes & Aydin (2020a)* | Adaptive De-Noising Overlapping Group Shrinkage Image Gradient Directional Masking Feature Extraction by MSER descriptor SVM Classification (non-linear Kernel) | 85.7 |
| | | Not applied | *Yuzkat, Ilhan & Aydin (2021)* | Data Augmentation Transfer Learning using CNN Architectures Decision Level Fusion | 90.73 |
| | | Not applied | *Ilhan & Serbes (2022)* | Data Augmentation Transfer Learning using VGG16 and GoogleNet (ImageNet) Decision Level Fusion (VGG16 + GoogleNet) | 90.87 |
| | | Not applied | *Mahali et al. (2023)* | Inappropriate Data Augmentation Transfer Learning SwinTransformer, MobileNetV3 and Auto encoder Fusion models (SwinMobile) | 91.7 |

(Continued)

| Data set name | Data set information | Orientation and cropping | Article | Method | Accuracy (%) |
|---|---|---|---|---|---|
| | | Not applied | *Aktas, Serbes & Osman Ilhan (2023)* | Data Augmentation Transfer Learning CNNs and Transformers | 89.43 |
| | | Not applied | Proposed study 2024 | Data Augmentation Transfer Learning ViTs, DEiTs and BEiTs | 92.5 |

- **Deployment environment:** Although transformer models can be computationally intensive, lightweight versions (*e.g.*, BEiT Tiny or distilled transformers) may be adapted for real-time deployment on standard GPU-enabled workstations typically available in hospital or fertility clinic settings.
- **Explainability and trust:** Visual attention maps generated by the transformer's attention mechanism can be employed to enhance explainability and increase clinicians' trust in the model's decisions.
- **Potential for research use:** The framework can assist researchers in large-scale morphology-based studies by automating the annotation process and enabling more objective statistical analysis of sperm defects.

These aspects position the proposed system not only as a strong candidate for clinical decision support but also as a reliable tool for academic and industrial research on male fertility.

## CONCLUSION

In this study, we proposed a fully automated transformer-based method for human sperm morphology classification, addressing key limitations of manual and semi-automated approaches. Leveraging the shape-awareness of vision transformers and optimized training configurations, our model achieved state-of-the-art accuracies of 93.52% on HuSHeM and 92.5% on SMIDS. Comparative analyses with CNN-based models, supported by Grad-CAM and Attention Maps, revealed that transformers focus on semantically meaningful, shape-related regions—particularly sperm heads—while CNNs tend to rely on texture or background cues. This highlights the superior capability of transformers in capturing morphology-critical features. The model operates without manual preprocessing, enabling integration into clinical or research workflows. Practical considerations such as efficiency, system compatibility, and real-time potential were also addressed. This work provides a foundation for deploying transformer-based solutions in fertility analysis, with future directions including explainable AI, model compression, and extension to motility and concentration assessment.

## LIMITATIONS AND FUTURE STUDIES

While the proposed transformer-based method shows promising results in sperm morphology classification, some limitations must be noted. Transformer models require

large datasets for generalization due to their high capacity and lack of CNN inductive biases. The SMIDS dataset was sufficient for stable training; however, the smaller HuSHeM dataset posed challenges. Data augmentation helped but may not fully capture real-world variability. Future work will explore advanced synthetic data generation such as generative adversarial networks (GANs) (*Goodfellow et al., 2020*) to enhance data diversity.

Regarding computational efficiency, transformer models trained faster than CNNs despite similar parameter counts, thanks to better parallelization. Thus, the notion that transformers are always computationally intensive does not fully apply here. Still, optimizing transformer training and inference on larger datasets will be considered in future studies.

Performance evaluation was rigorous, using $t$-tests on accuracy results from 11 models, two datasets, and 5-fold cross-validation. BEiT-Base achieved the best performance with 12 wins and eight draws out of 20 comparisons, validating its superiority.

Hybrid models combining CNNs and transformers were also explored to leverage their complementary strengths. However, the pure transformer-based BEiT-Base outperformed hybrids, suggesting its global feature extraction better suits sperm morphology classification. Future research will examine hybrid models on diverse datasets to identify scenarios where they may excel.

Future work includes more sophisticated fine-tuning strategies, such as the two-stage approach by *Ilhan & Serbes (2022)*, tailored for transformers. Domain-specific improvements like asymmetric patching and joint head-tail token fusion will be investigated to further boost performance.

BEiT's masked image modeling approach is well suited to medical imaging, evidenced by its strong sperm classification results (*Bao et al., 2021*). Emerging transformer variants with deformable or cross-scale attention may offer additional gains and will be evaluated.

To enable clinical deployment, runtime compression and optimization methods, including knowledge distillation, will be developed for GPU-limited environments. Dataset expansion through multi-center collaborations and GAN-based augmentation will improve generalization (*Goodfellow et al., 2020*).

Practical considerations for clinical adoption include integration with existing CASA software, hardware and infrastructure needs, compliance with privacy regulations, and ongoing software updates to maintain performance, security, and usability.

### Funding

This work was supported by the Turkey Health Institutes Presidency (TUSEB) under the project 'Mobile Phone-Based Sperm Motility Analysis and Reporting System' (Project No: 27698) and by the Scientific and Technological Research Council of Turkey (TUBITAK) ARDEB 1001 (Grant No: 122E164). The funders had no role in study design, data collection and analysis, decision to publish, or preparation of the manuscript.

## Grant Disclosures

The following grant information was disclosed by the authors:

Turkey Health Institutes Presidency (TUSEB): 27698.

Scientific and Technological Research Council of Turkey (TUBITAK) ARDEB 1001: 122E164.

## Competing Interests

The authors declare that they have no competing interests.

## Author Contributions

- Abdulsamet Aktas conceived and designed the experiments, performed the experiments, analyzed the data, performed the computation work, prepared figures and/or tables, and approved the final draft.
- Gorkem Serbes analyzed the data, prepared figures and/or tables, authored or reviewed drafts of the article, and approved the final draft.
- Hamza Osman Ilhan conceived and designed the experiments, performed the experiments, analyzed the data, performed the computation work, prepared figures and/or tables, and approved the final draft.

## Data Availability

The code is available in the Supplemental File.

The HuSHeM dataset is available at Mendeley: Shaker, Fariba (2018), "Human Sperm Head Morphology dataset (HuSHeM)", Mendeley Data, V3, DOI 10.17632/tt3yj2pf38.3.

The SMIDS dataset is available at Mendeley: ILHAN, Hamza (2022), "Sperm Morphology Image Data Set (SMIDS)", Mendeley Data, V1, DOI 10.17632/6xvdhc9fyb.1.

## Supplemental Information

Supplemental information for this article can be found online at http://dx.doi.org/10.7717/peerj-cs.3173#supplemental-information.

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
