# Peer review of "Unveiling the capabilities of vision transformers in sperm morphology analysis: a comparative evaluation"

_PeerJ Computer Science, doi:10.7717/peerj-cs.3173_

## Round 0.1 · original submission · Major Revisions

The reviewers have substantial concerns about this manuscript. The authors should provide point-to-point responses to address all the concerns and provide a revised manuscript with the revised parts being marked in different color.

·

Basic reporting

Introduction:
The introduction has managed to alert the reader about the particular area of study by very succinctly describing the context of the topic in question. A brief overview of the research area is presented along with the most significant issues or questions that need to be answered in this paper. The knowledge set provided is sufficient to ... as stated or claimed by this work related to the same. However, the motivation for the research is not presented sufficiently. Of course, it touches on the issues that emphasize the importance of the issue, but such explanations still lack the detail for the research to stand out in practical and theoretical terms. For example, identifying certain gaps in the existing literature or real-world use that would perhaps strengthen and practically modify the motivation behind the research.
The research question or hypothesis could perhaps be better formulated according to the purpose and scope of the paper, as such an outline comprehensively describes the objectives that the paper is trying to achieve. This will be useful in understanding what the paper is actually trying to achieve.
Formal Results:
The structure of the formal results section is improved, and the explanations for various terms and concepts are mostly clear. Some definitions, especially for more scientific or industry-based words, may need to be refined. This will help in making them more accessible to be understood across different levels of expertise.
Formal theorems and results are effectively explained, but other areas, especially aspects of evidence, require additional input. Key concepts are stated, but the proof is sometimes incomplete in its explanation for certain steps as well as assumptions and intermediate steps. This would not only help the reader understand each idea or thought behind the results obtained but also help remove the ambiguity that background approaches tend to have.
Suggestions for Improvement:
Motivation: Why this research is so important and what problems it addresses in previous works or practical areas need to be clearly explained to increase motivation.
Definitions: For some words, especially complex or specialized ideas, more accurate definitions should be provided to extend the understanding of this work to the general population.
Evidence: Extend the discussion of evidence by providing more details, especially for detailed arguments and assumptions. With this development, it will aid comprehension and make the reader follow the cognitive sequence of results in an orderly manner.
Overall, the introduction appropriately addresses the topic at hand but it would have been better if more explanation of the motivation was provided. The research results are mostly comprehensible but it would have been better if the dedication and evidence were more precise and clear to read.

Experimental design

Appropriateness to the Purpose and Scope of the Journal: The content of this article is appropriate for the purpose and scope of the journal for which it is intended. The topic is properly processed in accordance with the field of the journal, while the article appropriately targets an area that is appropriate for the type of articles the journal accepts. The author selects in a relevant manner the issues to be explored, thereby contributing to the advancement of science in the focus area.
Rigorous Investigation and High Technical & Ethical Standards: The research is rigorous in the conduct of individual investigations by placing extensive focus on the technical details required in activities such as experiments or analysis. The authors fully detailed their methodology including the selection of models, categories, algorithms, interactions, comprehensiveness, technology, and other methods applied. There is no indication of ethical oversight in this study, and the authors appear to follow ethical standards appropriate to the discipline.
Detailed Methods:
The methods used in this study are described in sufficient detail. However, to enable other researchers to replicate this study, there are sections that need to be supplemented with more details, such as the source code, the datasets used, and the computing infrastructure used. Providing scripts for reproduction of the results or links to related repositories would greatly help readers of this paper to reproduce these experiments in an easier way. This will increase the transparency of the research and further strengthen the validity of the results obtained.
Discussion on Data Pre-Processing:
There was a discussion on the data pre-processing done for this study, but the explanation could have been more detailed. Readers would appreciate the reasoning of the decisions made at this stage if the authors provided more details about the pre-processing methods applied, such as data cleaning and normalization or feature transformation used. This is especially useful as proper data processing can greatly affect the outcome of the study.
Description of Evaluation Methods, Assessment Metrics, and Model Selection:
The evaluation methods in this study have been adequately described, along with relevant metrics that are appropriate for the type of research being conducted. Although I think the metrics are sufficient to assess the performance of the implemented model. However, there is room for improvement here in terms of explaining the rationale for selecting a particular model. Why a particular model was chosen, many other simpler or complex models have been chosen. Supporting the model selection with justifiable arguments will definitely contribute positively to the overall discussion on the research topic.
Source Citation:
It can be inferred that the author has referred to relevant and current sources satisfactorily. Both direct quotes and paraphrases are placed correctly and in accordance with academic norms in the industry. However, there are some cases where there is a lack of mainly additional references that can strengthen the arguments or theories expressed in the article. There is a need to expand the literature referenced especially the recent ones to increase the scope of the theoretical framework used.
Suggestions for Improvement:
Methodological Details: Include more technical information such as source code, datasets, and reproduction scripts so that the study can be easily replicated by other researchers.
Data Pre-Processing: Explain more about the data pre-processing techniques used, such as normalization or handling of missing values to provide the audience with a better understanding of the issue under discussion.
Model Justification: Explain in more detail the rationale for the model selection and the assessment of other models to deepen the justification for the research model.
Additional References: Include more reference citations especially from recent articles to enhance the theoretical and methodological basis of the article.
Conclusion:
The article has done well in achieving the aims and objectives set by the journal, presenting an analysis that is appropriately conducted and that adheres to the technical and ethical standards of the profession. The articles are adequate, but it has been shown that they can be made more informative by advancing the discussion regarding the methods used, data processing, and model selection rationales.

Validity of the findings

Impact and Novelty:
The impact and novelty of the findings in this study were not explicitly evaluated in the article. Although the study makes a significant contribution to the field, it would have been better if the authors had gone into more depth about how the findings fill a gap in the existing literature or have a broader impact. A more detailed assessment of the novelty of the research results and their impact on the development of the field would enrich the discussion and provide a stronger context for the reader. In addition, meaningful replication should be encouraged, providing an explanation of the benefits of replicating the research in the future, as well as the reasons why it is important to continue or expand the research.
Well-Stated Conclusions:
The conclusions in this study are clearly stated and appropriate to the results obtained. The author successfully summarizes the main findings that support the objectives and hypotheses proposed in the introduction. However, to strengthen the conclusion, the author could have further emphasized how the findings directly relate to the development of science or practical applications in the field.
Experimentation and Evaluation:
The experiments and evaluations conducted in this study were well done and adequate. The evaluation methods applied were sufficient to assess the results of the experiments conducted. However, there are some areas that could have been further deepened, for example by testing multiple parameter variations or expanding the data sets used in the experiments to ensure more robust and reliable results. While the experiments are adequate, there is room to conduct a more comprehensive evaluation by considering more factors or using more diverse methods.
Well-Developed Arguments:
The argument built in this study is quite strong and clear, and effectively supports the objectives set out in the introduction. The author has successfully demonstrated how the study meets the desired objectives, providing evidence that supports the claims made. Nonetheless, to strengthen this argument, the author could have added more references from previous studies or compared the results with other relevant studies to demonstrate the validity and strength of the findings produced.
Identification of Unanswered Questions, Limitations, and Future Research Directions:
The conclusion is generally good, but it does not fully identify open questions, limitations, or directions for future research. Mentioning some of the limitations of the study, such as limited data, certain assumptions made, or other factors that could have affected the results, would have strengthened the conclusions and provided direction for further research. In addition, the direction for future research should be stated more clearly, giving the reader an idea of how this topic could be further developed in the future.
Suggestions for Improvement:
Impact and Novelty Assessment: Add a more in-depth discussion of the impact and novelty of the findings, as well as the contribution of the research to the field.
Research Replication: Encourage replication of the study by providing a clear explanation of the benefits of replication and the reasons why it is important to retest or expand the study.
Limitations and Future Research Directions: More clearly identify the limitations of the study as well as directions for future research. This will provide additional perspectives and open up opportunities for further research.
More Comprehensive Evaluation: Consider expanding the experiment by using a greater variety of data or more diverse evaluation methods to ensure more valid results.
Conclusion:
Overall, the findings in this study are valid and quite insightful, although the impact and novelty of the results have not been explored in detail. The conclusions are sound and support the research results, but there needs to be more explanation of limitations, unanswered questions and potential future research directions. The research fulfilled its purpose, but there is still room to deepen the analysis and identify next steps in this area.

Additional comments

This study demonstrates solid methodology and relevant findings, however there are some areas that could be improved to enhance its quality and impact.
Impact and Novelty: There needs to be more explanation regarding the impact of the findings, as well as how this research fills a gap in the existing literature or influences practice in the field. The novelty of the research has not been explored in depth.
Methodology and Evaluation: The methods used are self-explanatory, but the experiment could be expanded with more data variation or additional parameter tests to improve the validity of the results. A more in-depth evaluation is needed.
Conclusions and Future Research Directions: The conclusion of the study is good, but it is necessary to add the identification of the limitations of the study and the direction of future research. This will provide further perspective for the reader.
Overall, the research is valid and makes a valuable contribution, but the addition of an analysis of the novelty, limitations, and future research directions would strengthen the quality of this paper.

Reviewer 2 ·

Basic reporting

The paper introduces a significant amount of space to the historical development of deep learning and transformers. However, while some context is necessary, much of the content is not helpful for our reader since they may have the prior knowledge on deep learning-based classification task. Thus, I suggest the authors focus on sperm morphology analysis and why ViTs are a promising approach.

For related work in the introduction section, the authors could group them to reveal common findings or present them at a high level, followed by detailed explanations of each, instead of introducing each in excessive detail.

The figures are cluttered and difficult to interpret. Additionally, the axes and legends in the graphs lack clarity and should be reworked for better readability.

Experimental design

The paper compares transformer architectures against CNN baselines but lacks comparisons. Similar to the hybrid methods, i.e. CNN + ViT. Without such comparisons, the claims of superiority are not fully substantiated as a comparative study, even though some of these are mentioned in the introduction section.

I suggest including comparisons with state-of-the-art CNN and hybrid methods.

Besides this, the compared vision transformers are not recent (<2 years). Please consider including the latest ones.

The paper mentions the use of various augmentation scales but does not conduct a thorough ablation study to demonstrate the impact of augmentation techniques on performance. Please include an ablation study that evaluates the effects of augmentation.

Validity of the findings

The conclusions claim that ViT architectures are superior to CNNs for sperm morphology analysis, but the evidence is not comprehensive.

Additional comments

The dataset size might be small.

The impact of this work on clinical settings is not well discussed.

Reviewer 3 ·

Basic reporting

The manuscript employs clear and professional English throughout. While the introduction provides adequate context and literature review of sperm morphology analysis, the discussion of transformers in medical imaging could be expanded:

1.The literature review would benefit from a broader perspective on transformer applications in medical imaging.
2.While the current review thoroughly covers sperm morphology analysis applications, acknowledging these broader developments in medical image transformers would strengthen the paper's positioning and help readers understand how this work fits into the larger research landscape.

The paper has some structural and presentation issues:
1.Some implementation details are scattered across different sections, making methodology harder to follow
2.Figures (particularly Figure 2 and Figure 4) need higher resolution and better labeling
3. Mathematical notations should be consistently formatted throughout

Experimental design

The experimental methodology shows several limitations:

1.The current approach primarily applies existing transformer architectures without domain-specific modifications
2.The hyperparameter analysis, while thorough, focuses mainly on standard parameters rather than exploring domain-specific optimizations
3.The evaluation metrics should be expanded to include clinical relevance metrics beyond just classification accuracy
4.The comparison with existing methods needs more comprehensive statistical analysis to validate the claimed improvements

Validity of the findings

1.The performance improvements, while present, are incremental rather than transformative
2.Limited analysis of why transformer-based approaches perform better for this specific task
3.Insufficient discussion of clinical applicability and practical implementation challenges
4.Lack of ablation studies to understand the contribution of different components

The conclusions need to be better supported with:
- More detailed analysis of model behavior
- Clearer connection to clinical requirements
- Comprehensive error analysis
- Discussion of practical deployment considerations

Additional comments

The paper requires major revisions to enhance its contribution:
Technical aspects:
1.Develop novel architectural modifications specific to sperm morphology analysis
2.Incorporate medical imaging-specific attention mechanisms
3.Consider efficiency optimization techniques
4.Add comprehensive ablation studies

Theoretical aspects:
1.Deeper analysis of transformer limitations in medical imaging
2.More thorough investigation of model behavior
3.Better connection to clinical requirements
4.Extended discussion of practical implementation

·

Basic reporting

The manuscript is clear and professional but would benefit from enhanced clarity in some sections and additional contextual background.

Experimental design

The study is rigorous and well-structured but needs more explanation of parameter selection, dataset details, and baseline implementation.

Validity of the findings

The conclusions are supported by robust results, but further statistical analysis and discussion of limitations would strengthen the validity.

Additional comments

Thanks for nice article. Refer below for review comments.

Abstract:

1. The abstract is detailed but can be refined for better clarity. Consider revising the sentence, "Deep learning architectures have enabled the creation of systems that obtain sperm morphology features in a fully automated manner without requiring any pre-processing," to make it more concise.
2. Add a one-line summary of the significance of using Vision Transformers (ViT) over CNNs.
3. What would be the contribution to sperm morphology analysis compared to existing methods?

Introduction:
1. Expand on the challenges faced in sperm morphology analysis to provide a more comprehensive background for readers unfamiliar with the domain.
2. Include a brief discussion on why Vision Transformers represent an improvement over CNNs for this task.
3. Explicitly state the knowledge gap being addressed. For example: "While CNNs have shown promise in sperm morphology analysis, their reliance on manual pre-processing limits scalability. This work bridges this gap by leveraging Vision Transformers."

Experimental Setup:
1. Can you provide more details about the datasets (HuSHeM and SMIDS), including the number of images, resolution, and any preprocessing performed.
2. Include a justification for why these datasets are representative of real-world conditions.
3. You may need to explain the rationale for choosing specific hyperparameters, such as the learning rate and optimization algorithm, to enhance reproducibility.


Methodology:
1. The discussion of Vision Transformer architectures needs more depth. Include an explanation of why specific ViT architectures were chosen and how they differ from one another.
2. Elaborate on the metrics used for evaluation (e.g., accuracy) and explain why they are appropriate for this analysis.





Results:
1. Can you add a detailed discussion on why ViTs outperform CNNs in this context. Mention aspects like feature extraction, attention mechanisms, or robustness to data variability.
2. Include statistical validation (e.g., p-values or confidence intervals) to reinforce the significance of the findings.


Discussion:
1. Expand on how the proposed approach can be adopted in clinical or research settings. For instance, discuss its potential integration into automated sperm analysis tools.
2. What are the limitations - such as the size of datasets or the computational requirements of Vision Transformers, and suggest future directions to address these issues.

·

Basic reporting

The article is mostly clear and written in professional English

The introduction provides sufficient context regarding sperm morphology analysis and its significance in infertility treatment.

However:
The transition to the application of Vision Transformers (ViTs) in this field feels abrupt.
Suggestion: Expand on why ViTs, as opposed to CNNs, are particularly suitable for this application (e.g., their ability to capture long-range dependencies, focus on shape over texture, etc.).

Literature references are comprehensive and relevant, covering CNNs, ViTs, and their applications in sperm morphology analysis. The article should explicitly highlight gaps in the current literature to better justify the study.

Sections are well-organized, and the logical flow is maintained.

The "observer variability problem" (Line 44) could benefit from further elaboration, as it plays a significant role in motivating the automation of sperm analysis.

Mathematical or algorithmic aspects are not deeply covered, which is appropriate for this article's focus.

Experimental design

The article aligns well with the journal's focus on computer science and applied machine learning, specifically in biomedical applications. The study contributes meaningfully to the automation of sperm morphology analysis using Vision Transformers (ViTs).

The study provides sufficient details about the datasets, including their size, composition, and characteristics

While hyperparameter tuning is discussed, the exact methodology for selecting hyperparameters (e.g., grid search or random search) and their impact on performance should be elaborated further.

Suggestion: Include a brief section on how hyperparameter configurations were determined and whether alternative configurations were tested.

Conduct experiments using at least 3–5 different random seeds and report the mean and standard deviation of the results. This will help demonstrate the stability and reproducibility of the proposed approach. Add error bars to graphs (Figures 7, 8, and 9) to show model variability. Perform statistical tests (e.g., t-test) to confirm the significance of reported accuracy

Validity of the findings

The study demonstrates the effectiveness of Vision Transformers (ViTs) in fully automated sperm morphology analysis, eliminating the need for manual preprocessing (e.g., rotation, cropping).

The work systematically evaluates eight different ViT architectures, providing a comprehensive benchmarking study that has not been done before in this domain.

The comparative analysis with CNN-based models highlights ViTs' superior performance in recognizing sperm morphology based on shape rather than texture, aligning with the biological significance of morphology classification.

Add Visual and Computational Insights:
Include attention heatmaps to explain why ViTs work better than CNNs.
Compare training and inference time for ViTs vs. CNNs.

Reviewer 6 ·

Basic reporting

The authors have done an interesting work but settings aspects can be improved.

1. The Sections and sub-sections are not well-presented
2. The conclusion should be presented as a separate Section
3, Section 2 is entitled Experimental Results but unfortunately no results have been presented in the Section
3. Sections 2 and 2.1 can be combined and titled as Experimental Settings.
4. Authors have mentioned that they used data augmentation but have failed to mention the exact data augmentation techniques used in the study.
5. The introduction Section should be Section 1

Experimental design

The experimental setup in this study is ok. The authors have used contemporary deep-learning tools but have failed to present them properly. The reporting style of the experimental settings should be improved

Validity of the findings

The results are valid

Additional comments

The manuscript must be checked for grammatical errors

---

## Round 0.2 · accepted · Accept

The authors have well addressed all the concerns and I recommend accepting this manuscript.

·

Basic reporting

This version looks better, thanks for addressing the review comments.

Experimental design

This version looks better, thanks for addressing the review comments.

Validity of the findings

This version looks better, thanks for addressing the review comments.

Additional comments

This version looks better, thanks for addressing the review comments.

Reviewer 7 ·

Basic reporting

The authors have adequately addressed the concerns raised by previous reviewers. Specifically, 1) the authors have elaborated more on the motivations and impact of the proposed study; 2) they have included more recent references and justifications on the selected model; 3) they expanded in the discussion the limitations and future direction including the potential clinical applications; 4) they reorganized the tables and figures to improve the clarity of the paper.

Experimental design

The experimental design is appropriate as supported by other reviewer's comments. In addition, the authors have included more details on experimental settings and the source code to improve the transparency and reproducibility of the work. Additional comparisons with more recent state-of-the-art CNN and hybrid methods have been included by the authors to improve the comprehensiveness of current study. More analysis of ablation studies and model behavior have been included in the revised manuscript.

Validity of the findings

The results and conclusions are valid as supported by other reviewer's comment. The impact is assessed and the evaluations performed properly. The concerns have been properly addressed in the current version. The limitations and future direction has been further expanded in the revised manuscript. The error analysis with updated five-fold confusion-matrix metrics and statistical testing has been supplemented.

Additional comments

Please make comments on if the model is stable when different random seeds are used.

Reviewer 8 ·

Basic reporting

The comment here is to address the revised version of the paper. According to the previous comments and authors answers. I believe the authors have addressed the suggestions and made adequate revisions.

1. The authors have included clear motivation and background in the introduction. Additionally, they addressed the clinical relevance and limitations of prior models.

2. Introduction also now includes a clear and well-organized contribution list.

3. It was mentioned previously there were lack of explanations and definitions for terminologies. With the newer version, the authors included clear explanations such as ViT and CNN problems.

No additional comment

Experimental design

In terms of the experimental design. The major concerns are 1) the model selection and comparison to hybrid models, 2) lack of recent and adequate citations, 3) lack of clear methodology for reproducibility. From the newer version of the paper, the authors addressed the concerns properly.

1. the authors included the justifications of model selection. Additionally, two reference and hybrid models are now provided in the section 4.3 to further justify the model comparison.

2. there are more recent citations and additional citations included for scientific justifications in the introduction and throughout the paper. New citations provide strong evidence to support the claims made in the paper.

3. the authors improved on revising the experimental setup with more details, and provided clear instructions and availability of data sources.

No additional comment

Validity of the findings

In terms of validity of the findings, previous reviewers mentioned that 1) novelty and impact of the results, 2) data diversity and replication benefits, 3) limitations and future directions should be addressed. The authors addressed the concerns well and included additional evidence and revisions.

1. Again the bullet points in the introduction section included reasons and justifications for novelty and impact of the research. Furthermore, conclusions well-summarized the work in the paper and potential impact of the transformer-based solutions.

2. Additional replication benefits are included int he Data Availability and Conclusion sections

3. The authors added Limitations and Future Work section

No additional comment